# Large-scale multitrait genome-wide association analyses identify hundreds of glaucoma risk loci

Glaucoma, a leading cause of irreversible blindness, is a highly heritable human disease. Previous genome-wide association studies have identified over 100 loci for the most common form, primary open-angle glaucoma. Two key glaucoma-associated traits also show high heritability: intraocular pressure and optic nerve head excavation damage quantified as the vertical cup-to-disc ratio. Here, since much of glaucoma heritability remains unexplained, we conducted a large-scale multitrait genome-wide association study in participants of European ancestry combining primary open-angle glaucoma and its two associated traits (total sample size over 600,000) to substantially improve genetic discovery power (263 loci). We further increased our power by then employing a multiancestry approach, which increased the number of independent risk loci to 312, with the vast majority replicating in a large independent cohort from 23andMe, Inc. (total sample size over 2.8 million; 296 loci replicated at $P < 0.05$, 240 after Bonferroni correction). Leveraging multiomics datasets, we identified many potential druggable genes, including neuro-protection targets likely to act via the optic nerve, a key advance for glaucoma because all existing drugs only target intraocular pressure. We further used Mendelian randomization and genetic correlation-based approaches to identify novel links to other complex traits, including immune-related diseases such as multiple sclerosis and systemic lupus erythematosus.

Primary open-angle glaucoma (POAG) is a leading cause of irreversible blindness world-wide[1,2]. It is often asymptomatic until later stages, causing optic nerve damage manifested by cupping and visual field loss[3]. Large vertical cup-to-disc ratio (VCDR) and elevated intraocular pressure (IOP) are two key POAG endophenotypes[4]. POAG is one of the most heritable common diseases[5], with previous genome-wide association studies (GWASs) identifying 127 loci, collectively explaining 9.4% of the familial risk[6]. However, these loci only account for a moderate fraction of the heritability, many risk loci have not been discovered and their biological functions remain largely unknown.

Multitrait methods have demonstrated substantial improvements in power for uncovering novel genetic loci when incorporating data from related endophenotypes[7]. Both VCDR and IOP are highly genetically correlated with glaucoma (genetic correlation 0.50 [s.e.m. = 0.05] and 0.71 [s.e.m. = 0.04], respectively)[8]. In recent years, the number of samples with IOP and VCDR measurements has significantly increased. For instance, advances in artificial intelligence (AI) provided a new opportunity for accurate phenotyping of the optic nerve head, leading to an increased number of samples with VCDR[9]. In our previous AI-based GWAS, we predicted VCDR values for 282,100 fundus images based on convolutional neural network (CNN) models[9]. In parallel, IOP measurements

✉e-mail: hanxikun2017@gmail.com

**Fig. 1 | Study design.** MGB Biobank, Mass General Brigham Biobank; DD, disc diameter; LDSC, linkage disequilibrium score regression.

are also available from multiple large-scale biobanks ($n > 150,000$). This information greatly expands the effective sample size for glaucoma in a multitrait framework and substantially enhances power for glaucoma gene discovery. Moreover, as VCDR and IOP are not strongly correlated with each other (genetic correlation 0.22 [s.e.m. = 0.03])[8], a large-scale analysis has the potential to uncover distinct genetic signals from IOP and VCDR; VCDR signals are particularly interesting as these may help uncover putative 'neuro-protection' drug candidates.

Herein, leveraging new and existing genetic data for POAG, VCDR and IOP, we perform a large-scale multitrait analysis of GWAS (MTAG) to identify novel POAG loci. We integrate data across different ancestries to aid in fine mapping. We utilize a range of omics datasets to improve our understanding of the underlying biological mechanisms for POAG, leading to improved druggable target discovery for this blinding disease. We also exploit the very large effective sample size to conduct genetic causal inference analysis to assess the relationships between a wide range of complex diseases/traits and POAG susceptibility.

## Results

### Study design

The study design is illustrated in Fig. 1 (see also Supplementary Table 1). We first performed an MTAG in the European ancestry population, including GWASs for POAG (29,241 cases and 350,181 controls) and its two key endophenotypes, VCDR ($n = 111,724$) and IOP ($n = 153,604$). The identified novel POAG loci from MTAG were then replicated in a large-scale independent glaucoma GWAS (23andMe, Inc. study, 84,910 cases and 2,736,075 controls). We further conducted a multiancestry meta-analysis for POAG, combining the MTAG POAG output from the European ancestry population and samples from Asian (6,935 cases and 39,588 controls) and African (3,281 cases and 2,791 controls) ancestry populations. We applied a variety of fine-mapping and post-GWAS functional analytical approaches to prioritize the genetic findings, identify druggable targets and characterize potential biological mechanisms underlying POAG.

**POAG MTAG analysis of European ancestry population identifies 263 loci.** In the MTAG analysis combining GWASs of POAG, VCDR and IOP in the European ancestry population, we identified 263 independent loci for POAG; 81 loci were novel (not within ±500 kilobases (kb) of previously known loci; Fig. 2a, Supplementary Table 2, Extended Data Figs. 1–3 and Supplementary Data 1). The proportion of the familial risk explained by the genome-wide significant independent SNPs was 14.1%. This represents a 50% increase over and above the previously reported estimate (9.4%), based on a previous meta-analysis that identified 127 SNPs[6]. The 81 completely novel loci (not within ±500 kb of previously known loci) contributed 2.5%, with the remainder of the difference (2.2%) attributable to additional independent SNPs within previously reported loci. The identified lead SNPs were then replicated in an independent cohort using 23andMe (261 SNPs available in the 23andMe study): 60% of SNPs ($n = 156$) passed the genome-wide significance level ($P < 5 \times 10^{-8}$) in the 23andMe study, 85% of SNPs ($n = 223$) were significant after Bonferroni correction ($P < 0.00019$) and 98% of SNPs ($n = 256$) reached a nominal significance level ($P < 0.05$). We found a very high concordance of the effect sizes between the MTAG discovery and the 23andMe replication (Pearson's coefficient 0.97, $P = 5.99 \times 10^{-154}$; Fig. 3a and Extended Data Figs. 1 and 4). In the MTAG analysis, the maximum false discovery rate (FDR) for POAG was 0.004, suggesting no evidence of inflation due to violation of the homogeneity assumption in the MTAG analysis.

The top ten novel loci included SNPs located in or near *FOXF1*, *CTNND1*, *FENDRR*, *GNB3*, *FLNB*, *COL8A1*, *SLC30A10*, *VAV2*, *MYO16* and *HSPA12A* (Supplementary Table 2). *FOXF1* is a forkhead transcription factor gene on 6p25, disruption of which has been reported to cause a range of ocular developmental abnormalities associated with glaucoma[10,11]. The lead SNP rs1728414 of *FOXF1* was associated with POAG at $P = 1.45 \times 10^{-6}$ in the previous POAG GWAS[6], and reached $P = 1.97 \times 10^{-18}$ in the current MTAG POAG GWAS. This SNP also reached genome-wide significance level in our 23andMe replication dataset ($P = 2.16 \times 10^{-25}$), confirming that our MTAG approach can identify novel POAG loci when the effective sample size was dramatically increased.

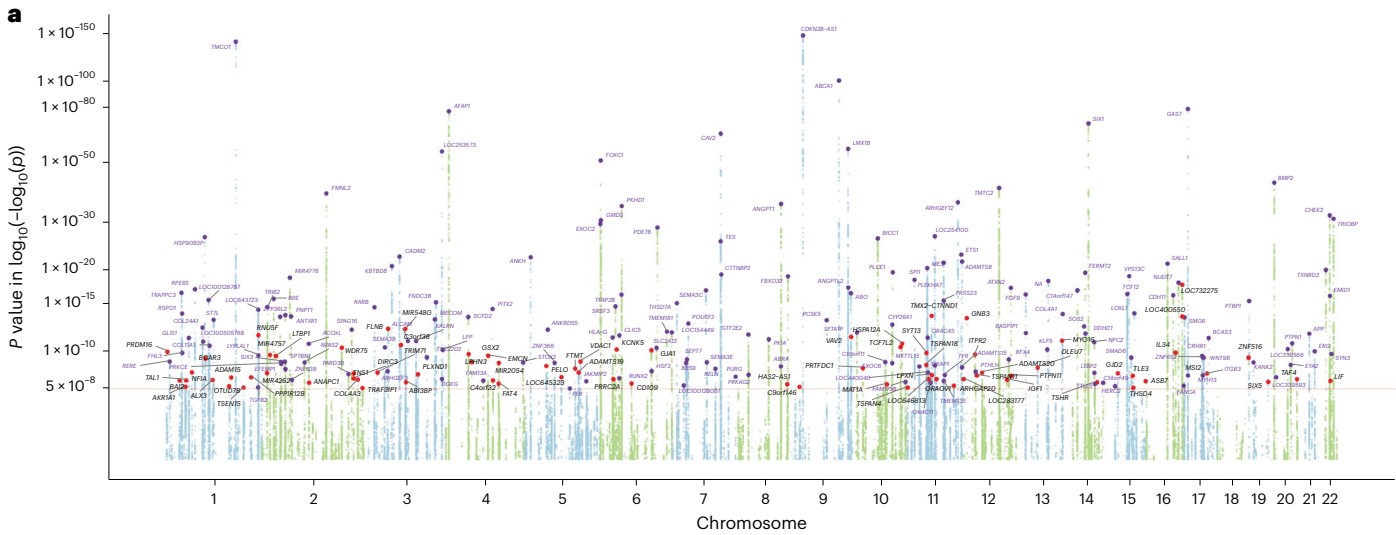

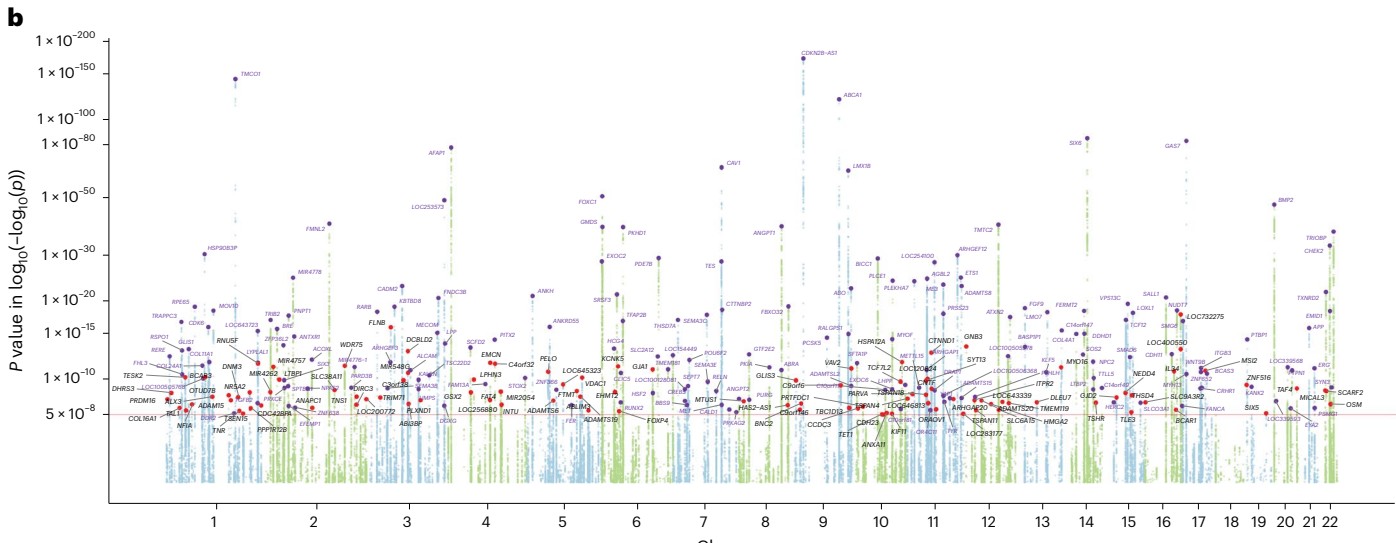

**Fig. 2 | Manhattan plots displaying POAG GWAS P values. a**, Plot shows 263 loci from POAG MTAG in the European ancestry population. **b**, Plot shows 312 loci from POAG multiancestry meta-analysis. In these plots, the *y* axis shows the *P* values of SNPs in log–log scale. The red horizontal line is the genome-wide significance level at $P = 5 \times 10^{-8}$. SNPs with $P < 1 \times 10^{-4}$ are not shown in the plots. Previously unknown loci are highlighted with red dots, and the nearest gene names are in black text. Known SNPs are highlighted with purple dots, and the nearest gene names are in purple text. All tests were two-sided.

The identified POAG loci from the European ancestry population were compared with POAG GWASs from Asian and African ancestry populations. The Pearson's coefficient was 0.77 ($P = 6.9 \times 10^{-45}$) and 0.507 ($P = 3.8 \times 10^{-17}$) in Asian and African ancestry populations, respectively (Fig. 3c,d), suggesting moderately high concordance across different ancestries.

**Multiancestry meta-analysis identifies 312 POAG loci.** We then conducted a multiancestry meta-analysis using the POAG MTAG output in the European ancestry population and POAG GWASs from Asian and African ancestry populations. In total, we identified 312 independent loci; 109 loci were novel (Fig. 2b, Supplementary Table 3, Extended Data Fig. 5 and Supplementary Data 2). We further replicated the loci from multiancestry meta-analysis in the 23andMe study (302 SNPs available in 23andMe): 169 SNPs (56%) passed the genome-wide significance level ($P < 5 \times 10^{-8}$) in the 23andMe study, 240 SNPs (79%) passed Bonferroni correction ($P < 0.00017$) and 296 SNPs (98%) reached a nominal significance level ($P < 0.05$). Overall, we found a high concordance of effect sizes between the multiancestry meta-analysis and the 23andMe replication dataset (Pearson's coefficient 0.96, $P = 1.22 \times 10^{-164}$; Fig. 3b). Many of the novel loci represent druggable targets (described further below in 'Prioritization of drug targets for POAG' section).

**Comparison with rare variant association results from exome sequencing.** We compared the identified common variant POAG loci from our GWAS approach with rare variant association analysis from exome sequencing[12]. We identified a common variant rs281857 near *OPTN* associated with POAG. *OPTN* harbors several well-known high-penetrance glaucoma risk variants[13]. The variant rs281857 was replicated in the 23andMe cohort ($P = 1.97 \times 10^{-7}$) and had a small but detectable effect on both IOP ($P = 0.0052$) and VCDR ($P = 0.0036$); rs281857 is in linkage equilibrium with both the rare Mendelian disease variants and the reported common variant (rs11258194) in the report by Rezaie et al.[13]. We found no links between rs281857 and *OPTN* expression, and the specific mechanisms for how rs281857 alters glaucoma risk are unclear. Other significant genes identified in rare variant gene-based association analysis included *FKBP9*, *LTBP2*, *COL2A1* and the well-known glaucoma gene *MYOC*. We found that *FKBP9* is 400 kb from

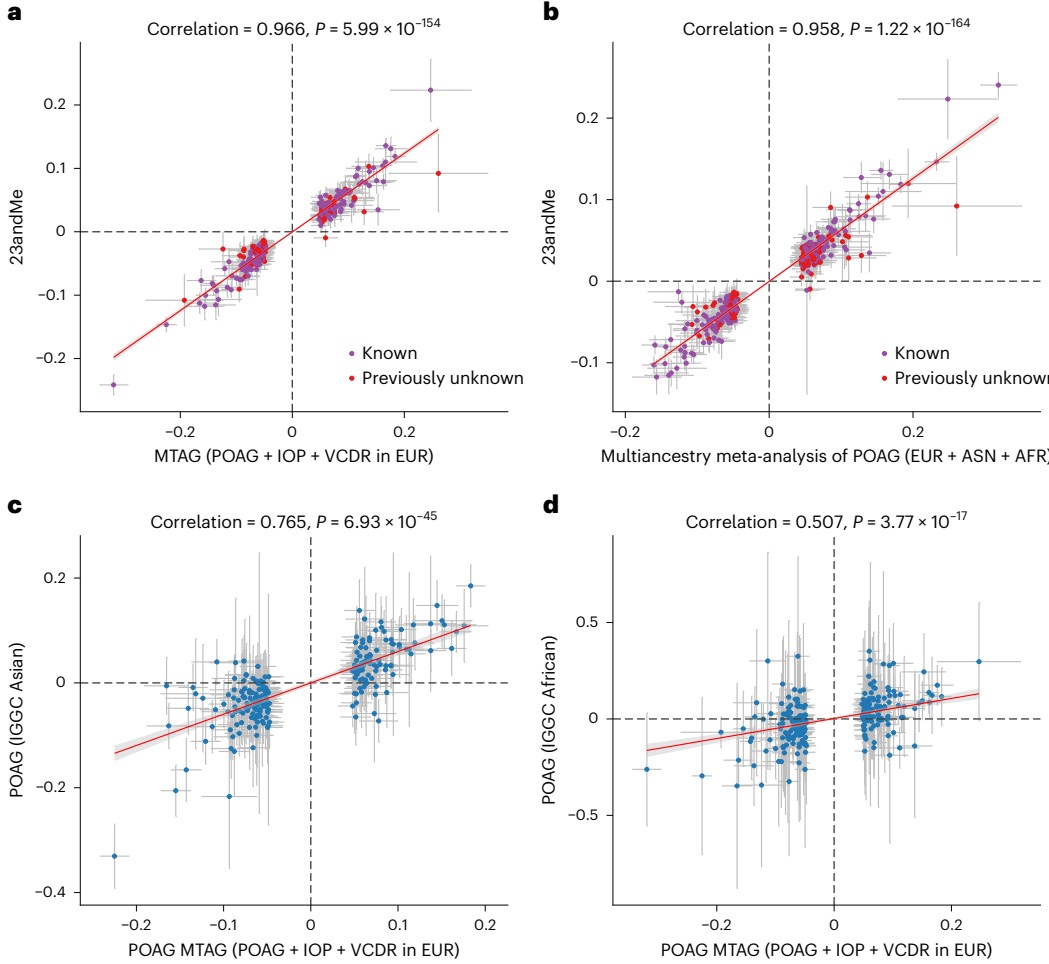

**Fig. 3 | Comparison of the effect sizes for genome-wide significant independent SNPs. a,** Plot shows 261 (two SNPs were unavailable in 23andMe) genome-wide significant independent SNPs identified from POAG MTAG in the European population versus glaucoma GWAS in 23andMe. The Pearson's coefficient is 0.966 ($P = 5.99 \times 10^{-154}$, $n = 261$ independent SNPs). **b,** Plot shows effect sizes for 302 (10 SNPs were unavailable in 23andMe) genome-wide significant independent SNPs identified from the POAG multiancestry

meta-analysis versus glaucoma GWAS in 23andMe. The Pearson's coefficient is 0.958 ($P = 1.22 \times 10^{-164}$, $n = 302$ independent SNPs). In **a** and **b**, previously unknown SNPs are colored in red. **c,d,** Plots ($n = 261$ independent SNPs) show POAG MTAG from the European ancestry population versus POAG GWAS from Asian (**c**) and African (**d**) ancestry populations. The dots show the effect sizes of SNPs, and the error bars show the 95% confidence interval of the estimations of SNP effect sizes. AFR, African ancestry; ASN, Asian ancestry; EUR, European ancestry.

the common SNP rs1544557 in the POAG MTAG GWAS, *LTBP2* is 20 kb from the common POAG SNP rs754634 and *COL2A1* is associated with VCDR (lead SNP rs12821310)[8]. Our findings are in keeping with other complex diseases where a gene can influence disease risk through both rare and common variants.

**Gene-based and pathway analysis.** The MTAG per-SNP results were used as input for gene-based analysis, identifying 355 significant genes for POAG after adjusting for multiple testing (Bonferroni correction, $P < 2.68 \times 10^{-6}$). Of the 355 significant genes, 304 genes were near the genome-wide significant SNPs (Supplementary Table 4). In the pathway analysis based on the gene-based results, we uncovered 32 pathways after Bonferroni correction ($P < 3.23 \times 10^{-6}$) (Supplementary Table 5). Implicated pathways included those involved in collagen formation, blood vessel development and cardiovascular system development.

**Classification of POAG loci into VCDR- or IOP-specific SNPs.** Based on a hierarchical clustering approach, we classified the 263 MTAG POAG loci of European ancestry into SNPs that were more likely to be associated with VCDR ($n = 92$) or IOP ($n = 171$) (Fig. 4a and Supplementary Table 2, column 'assign_SNP'). Overall, for the set of SNPs clustered as VCDR-specific SNPs, the effect of each SNP on VCDR showed a very

high concordance with the MTAG POAG effect size (Fig. 4b); the same was true for IOP-specific SNPs (Fig. 4c). The classification of VCDR- and IOP-specific SNPs from the clustering approach was consistent with a multitrait colocalization method, where posterior probability was used to support a colocalization of each POAG locus with VCDR or IOP (Extended Data Fig. 6 and Supplementary Table 6). The classified VCDR-specific SNPs were used to identify potential 'neuro-protection' drug targets (described further below in 'Prioritization of drug targets for POAG' section).

**Gene expression/alternative splicing quantitative trait locus colocalization analysis prioritizes causal genes in POAG loci.** Using the Bayesian colocalization method eCAVIAR, we tested whether any of the gene expression quantitative trait loci (eQTLs) and alternative splicing quantitative trait loci (sQTLs) in 49 Genotype-Tissue Expression (GTEx) tissues or retina tissue share one or more causal variants with the MTAG or multiancestry POAG loci. Target genes of the colocalizing eQTLs/sQTLs may be causal to POAG. We found significant colocalization with one or more eQTLs/sQTLs for 139 (52.9%) of the replicated MTAG POAG loci in European ancestry, 40 of which are novel loci (Supplementary Table 7), and for 148 (47.4%) of the replicated multiancestry loci, 38 of which are novel loci (Supplementary Table 8). The colocalization

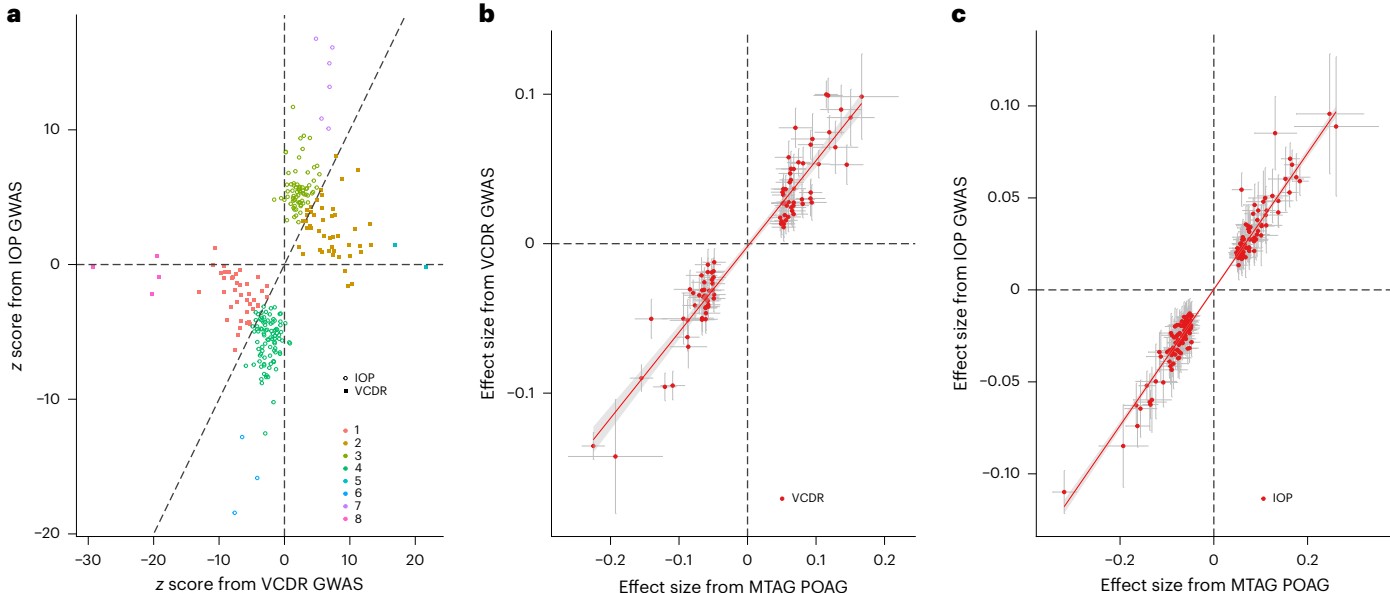

**Fig. 4 | Classification of POAG loci into VCDR- or IOP-specific SNPs. a**, Plot shows hierarchical clustering of 263 MTAG POAG loci (a multitrait colocalization approach is shown in Extended Data Fig. 6). Based on the z scores (dashed line shows y = x where the z scores of VCDR and IOP are equal), there is a subset of SNPs that act primarily via IOP (n = 171, dots) and a subset of SNPs acting primarily via VCDR (n = 92, squares). **b**, Plot displays the effect sizes on VCDR and on POAG for the 92 VCDR-specific SNPs. **c**, Plot displays the effect sizes on IOP and on POAG for the 171 IOP-specific SNPs. The dots show the effect sizes of SNPs, and the error bars show the 95% confidence interval of the estimations of SNP effect sizes.

analysis proposed on average 3.8 ± 0.55 causal genes per locus, with a single gene proposed for 22% of the loci (Supplementary Tables 7 and 8).

**Transcriptome-wide association study identifies genes associated with risk of POAG.** In the transcriptome-wide association study (TWAS) analysis using retinal tissue and MTAG POAG GWAS summary statistics in the European ancestry population, we identified 86 genes that were associated with POAG risk after Bonferroni correction (Supplementary Table 9). Of the 86 genes, 20 genes had no genome-wide significant SNPs within the gene (best GWAS SNP $P > 5 \times 10^{-8}$). From the TWAS analysis, we prioritized potential causal genes. For instance, *FOXF1* was again identified from TWAS as a strong candidate gene resulting from polymorphism at the top novel SNP rs1728414 (chr16:86393405).

**Mapping the effects of plasma proteome on risk of POAG.** Using the MTAG POAG GWAS summary statistics and large-scale protein quantitative trait locus (pQTL) associations of European ancestry, we performed Mendelian randomization (MR) and identified 33 proteins that were potentially causally associated with risk of POAG (FDR adjusted $P < 0.05$; Supplementary Table 10), including the protein encoded by *ENG* that was also identified in the TWAS analysis based on retinal tissue. Of the 33 identified proteins, 15 and 20 were also associated with VCDR and IOP (FDR adjusted $P < 0.05$), respectively (Supplementary Table 10). We then performed a proteome-wide association study (PWAS) using an independent proteomics dataset to train prediction models for genetically imputing proteins, and ten proteins passed Bonferroni correction ($P < 0.05/1308 = 3.79 \times 10^{-5}$; Supplementary Table 11).

**Prioritization of drug targets for POAG.** Leveraging multiple lines of genetic evidence (that is, genome-wide significant loci or gene-based results from MAGMA, TWAS significant genes based on eQTL data in retina, eQTL/sQTL colocalization in GTEx tissues or retina, and MR-supported putative causal proteins based on pQTL data in plasma), we identified 69 potential drug target genes for POAG (Supplementary Table 12). Of these, we prioritized 17 druggable genes with at least two levels of genetic evidence (Table 1); for example, *COL11A1* and *CYP26A1* were supported with proximity to the lead GWAS SNPs, TWAS in retina

and eQTL colocalization in several tissues; *NDUFS3* was supported with MAGMA gene-based test, TWAS in retina and eQTL colocalization in several tissues; and *ENG* was supported with TWAS in retina and pQTLs in plasma. Examples of the existing drugs targeting these genes include collagenase clostridium histolyticum and ocriplasmin (collagen hydrolytic enzymes) targeting *COL11A1*; talarozole, a cytochrome P450 26A1 inhibitor (the protein encoded by *CYP26A1*); metformin and ME-344 (mitochondrial complex I inhibitors) targeting *NDUFS3*; and carotuximab, an Endoglin inhibitor (the protein encoded by *ENG*). In addition, we highlighted several drugs targeting genes that increase the risk of POAG, most likely through affecting VCDR, independent of IOP (without an apparent effect on IOP; Table 1): for example, *CHEK2*, encoding a cell cycle regulator and putative tumor suppressor; *RPE65*, which encodes retinoid isomerohydrolase, a component of the vitamin A visual cycle in retinal rod and cone photoreceptors[14]; and *TNFSF13B*, which encodes a tumor necrosis factor ligand family. These VCDR genes are potential targets for development of neuroprotective treatments for POAG, which currently are not available (see more details in Discussion section).

**Traits genetically associated with POAG.** From bivariate genetic correlation analysis of 1,347 GWAS summary statistics for complex diseases or traits, we identified 24 traits that were genetically correlated with POAG, VCDR or IOP after FDR correction (Supplementary Table 13 and Extended Data Fig. 7). For example, cognitive performance, intelligence and education were positively correlated with VCDR. In our two-sample MR analysis, we identified 14 traits that showed putatively causal effects on risk of POAG (FDR $P < 0.05$; Supplementary Table 14), including multiple sclerosis, systemic lupus erythematosus, type 2 diabetes (T2D) and immune cells (Fig. 5). From colocalization analysis, we identified shared genetic regions between the associated traits and POAG (Supplementary Table 15). For instance, we identified one genomic region (gene *ATXN2*) with a high posterior probability (PP4 = 0.98) between systemic lupus erythematosus and POAG (Extended Data Fig. 8). We performed sensitivity analyses using different MR methods to evaluate the robustness of the MR findings. We observed no evidence of directional pleiotropy effects from the MR-Egger intercept (intercepts close to zero). Heterogeneity of outlier SNPs was tested using MR-PRESSO (MR pleiotropy residual

**Table 1 | Prioritized drug targets for POAG**

| Gene | Mapping criteria | Only VCDR effect[a] | Drug name | Mechanism of action | Diseases under trial |
|---|---|---|---|---|---|
| COL11A1 | Nearest gene/MAGMA, TWAS, eQTL coloc | No | Collagenase clostridium histolyticum, ocriplasmin | Collagen hydrolytic enzyme | Macular degeneration and macular holes, diabetic macular edema, retinal vein occlusion |
| CYP26A1 | Nearest gene, TWAS, eQTL coloc | Yes | Talarozole | Cytochrome P450 26A1 inhibitor | Acne, psoriasis, inflammation |
| NDUFS3 | MAGMA, TWAS, eQTL coloc | No | Metformin, ME-344 and others | Mitochondrial complex I (NADH dehydrogenase) inhibitor | Stargardt disease, muscular dystrophy, diabetic retinopathy, cognitive impairment, cardiovascular disease, mental disorders, cancer |
| ENG | TWAS, pQTL | No | Carotuximab | Endoglin inhibitor | Age-related macular degeneration, cancer |
| CHEK2 | Nearest gene/MAGMA, eQTL coloc | Yes | Prexasertib, XL-844 | Serine/threonine-protein kinase Chk2 inhibitor | Cancer |
| ANGPT1 | Nearest gene/MAGMA, eQTL coloc | No | Trebananib, AMG-780 | Angiopoietin-1 inhibitor | Cancer |
| PRKCE | Nearest gene/MAGMA, eQTL coloc | No | Midostaurin, KAI-1678 and others | Protein kinase C inhibitor, protein kinase C epsilon inhibitor | Systemic mastocytosis, cancer, pain, psoriasis, liver disease |
| COL4A1 | Nearest gene/MAGMA, sQTL coloc | No | Collagenase clostridium histolyticum, ocriplasmin | Collagen hydrolytic enzyme | Macular degeneration and macular holes, diabetic macular edema, retinal vein occlusion |
| F2 | MAGMA, pQTL | No | Bivalirudin, argatroban and others | Thrombin inhibitor | Cardiovascular diseases |
| COL5A2 | MAGMA, eQTL coloc | No | Collagenase clostridium histolyticum | Collagen hydrolytic enzyme | Macular degeneration and macular holes, diabetic macular edema, retinal vein occlusion, abnormality of connective tissue, stroke |
| ITGB5 | MAGMA, eQTL coloc | No | Cilengitide | Integrin alpha-V/beta-5 antagonist | Cancer, kidney disease, myelodysplastic syndrome |
| PSMC3 | MAGMA, eQTL coloc | No | Oprozomib | 26S proteosome inhibitor | Cancer |
| CRHR1 | MAGMA, eQTL coloc | No | SSR125543, verucerfont and others | Corticotropin releasing factor receptor 1 antagonist | Social anxiety disorder, irritable bowel syndrome, major depressive disorder, congenital adrenal hyperplasia |
| MAPT | MAGMA, eQTL/sQTL coloc | No | Gosuranemab, semorinemab and others | Microtubule-associated protein tau inhibitor | Alzheimer's disease, progressive supranuclear palsy |
| NR1H3 | MAGMA, eQTL/sQTL coloc | No | BMS-852927, hyodeoxycholic acid, RGX-104 | LXR-alpha modulator, LXR-alpha agonist, liver X receptor agonist | Hypercholesterolemia, neoplasm |
| TGFB3 | MAGMA, sQTL coloc | No | Luspatercept, bintrafusp alfa, fresolimumab | Transforming growth factor beta inhibitor | Myeloproliferative disorder, myelodysplastic syndrome, myelofibrosis, anemia, cancer |
| ITGB3 | Nearest gene, sQTL coloc | No | Abciximab, tirofiban and others | Integrin alpha-IIb/beta-3 inhibitor, integrin alpha-V/beta-3 antagonist | Cardiovascular diseases, psoriasis, cancer, COVID-19, anemia |
| HTR1F | eQTL coloc | Yes | Almotriptan malate, dexfenfluramine, amisulpride and others | Serotonin 1f (5-HT1f) receptor agonist, serotonin (5-HT) receptor agonist, serotonin (5-HT) receptor antagonist | Mental disorders, dementia, migraine disorder, kidney disease |
| PDE6C | eQTL coloc | Yes | Dipyridamole, pentoxifylline | 3',5'-cyclic phosphodiesterase inhibitor | Duchenne muscular dystrophy, diabetes, diseases of heart, kidney, and liver, cancer, anemia, mental disorders |
| RPE65 | Nearest gene | Yes | Emixustat, voretigene neparvovec | Retinoid isomerohydrolase inhibitor, retinoid isomerohydrolase positive modulator | Macular degeneration, diabetic retinopathy, Stargardt disease, retinal dystrophy, Leber congenital amaurosis |
| CDC7 | Nearest gene | Yes | BMS-863233, NMS-1116354 | Cell division cycle 7-related protein kinase inhibitor | Refractory hematologic cancer, neoplasm |
| TNFSF13B | Nearest gene | Yes | Belimumab, blisibimod and others | Tumor necrosis factor ligand superfamily member 13B inhibitor | Optic neuritis, multiple sclerosis, systemic lupus erythematosus and other immune-related diseases |

**Table 1 (continued) | Prioritized drug targets for POAG**

| Gene | Mapping criteria | Only VCDR effect[a] | Drug name | Mechanism of action | Diseases under trial |
|------|------------------|---------------------|-----------|---------------------|----------------------|
| *CD248* | TWAS | Yes | Ontuxizumab | Endosialin inhibitor | Soft tissue sarcoma, metastatic melanoma, neoplasm |
| *GSR* | TWAS | Yes | Carmustine, oxiglutatione | Glutathione reductase inhibitor, glutathione reductase | Neuromyelitis optica, abnormality of blood tissues, cancer, myelodysplastic syndrome |
| *LAMB2* | TWAS | Yes | Ocriplasmin | Laminin hydrolytic enzyme | Macular degeneration, macular holes, diabetic macular edema, retinal vein occlusion, uveitis, stroke, deep vein thrombosis |

This table presents the existing approved drugs that target genes whose effect on POAG is supported by at least two lines of genetic evidence (eQTL, pQTL or proximity to the most significant SNPs; *n* = 17), or genes (*n* = 8) that affect POAG most likely through VCDR, without an apparent effect on IOP. [a]For genes mapped based on pQTL support, VCDR genes are defined as those associated with VCDR (FDR ≤ 0.05), but not IOP, in the pQTL MR analyses (Supplementary Table 10). For genes mapped based on proximity to the most significant GWAS SNPs, VCDR genes are the nearest genes to the most significant SNPs that are predicted to affect both POAG and VCDR (but not IOP) with a posterior probability > 0.7 in colocalization analysis (Supplementary Table 6). For genes mapped based on TWAS and eQTL/sQTL colocalization, VCDR genes are those whose best corresponding GWAS SNPs are genome-wide significant for VCDR, but not associated with IOP (or only nominally associated with IOP).

sum and outlier), and the outlier-corrected results were essentially the same (Supplementary Table 16). We found no evidence of bidirectional effects for the identified traits (except the association between VCDR and optic disc area; Extended Data Fig. 9).

## Discussion

In this study, we performed a large-scale multitrait POAG GWAS identifying 263 loci in the single largest ancestry group (European ancestry). Additional cross-ancestry meta-analysis identified 312 loci, including 109 that were completely distinct from previously reported loci. Leveraging omics data and multiple levels of genetic evidence, we prioritized 69 putative drug targets for POAG (including 17 with at least two levels of supporting genetic evidence), with many linked to genetic loci that act at least in part directly via optic nerve head damage and not via raised IOP, making them promising 'neuro-protection' candidates. Finally, we systematically evaluated more than 1,000 publicly available GWAS summary statistics and identified several immune disorders that are possibly causally associated with POAG.

For genetically correlated traits, multitrait approaches have shown utility in boosting statistical power for detecting novel genetic associations[7,8]. To our knowledge, in the current study, we assembled the largest-scale genetic datasets to date for POAG, VCDR and IOP, and nearly tripled the number of POAG loci[6]. We strongly replicated most of the novel loci using a very large independent dataset (the 23andMe study). The high replication rate is likely due to both the increased power of MTAG to identify genuine POAG loci and the large sample size of the 23andMe study allowing for robust estimates for replication. The values of mean chi-square were 1.55 and 1.18 in the current MTAG POAG GWAS and the previous largest POAG GWAS from the International Glaucoma Genetic Consortium (IGGC) (with 16,677 POAG cases and 199,580 controls), respectively, indicating that we have tripled the effective sample size for POAG[6,7]. These results are in keeping with our recent modeling[15] showing that leveraging endophenotypes of POAG is an effective means to increase the statistical power to identify novel POAG loci.

Genetic studies have provided new insights to identify therapeutic targets. Drug mechanisms with genetic support are two times more likely to be approved than those without it[16,17]. Using multiple levels of genetic evidence, we prioritized 17 putative drug targets for POAG with at least two levels of genetic evidence in the current study. For example, carotuximab targets *ENG* (Endoglin), which we showed to be a potential causal gene for POAG based on the evidence from the integration of eQTL and pQTL data. Carotuximab is an Endoglin inhibitor that has been under consideration in clinical trials for treatment of exudative age-related macular degeneration[18], further highlighting its potential for treatment of neurodegenerative ocular diseases. In support of this, it has been shown that increased expression of Endoglin

results in retinal neovascularization and retinopathy in mice[19]. Our MR analysis predicting proteomic effects on POAG risk implicated 33 proteins, including TEK, a receptor from the protein tyrosine kinase Tie2 family. *TEK* gene mutations have previously been found in congenital glaucoma families with variable expressivity[20,21]. Genomic regions that include genes encoding TEK ligands ANGPT1 and ANGPT2 have previously been associated with both IOP and POAG[8,22]. Multiple drugs target TEK, including regorafenib, which was trialed for use in various diseases, including macular degeneration[23].

Current therapies for glaucoma are limited as they rely solely on reducing high IOP. Therefore, it is crucial to develop IOP-independent therapeutic strategies to counter neurodegeneration in glaucoma. To address this, as well as identifying genes such as *ENG* that likely act via IOP, we highlighted several drugs targeting genes that likely affect POAG through thwarting optic nerve damage, independent of IOP. In support of their possible neuroprotective effects, some of these drugs are in clinical trials for treatment of neurodegenerative ocular diseases such as retinal dystrophy, atrophic macular degeneration and macular holes (for example, drugs such as voretigene neparvovec, emixustat and ocriplasmin). In addition, some of these drugs are in clinical trials for treatment of neurodegenerative diseases of the central nervous system such as multiple sclerosis (for example, belimumab). For instance, the gene *TNFSF13B* is druggable by TNFSF13B inhibitors which include belimumab, blisibimod, tabalumab and atacicept. These drugs were trialed for use in optic neuritis, multiple sclerosis, systemic lupus erythematosus and other immune-related diseases. Of interest, we also observed a significant genetic correlation between POAG and immune-related diseases, including lupus and neurodegenerative conditions such as multiple sclerosis. Further studies to confirm the causality of these genes in vitro and in vivo may support the potential of repurposing these drugs as novel neuroprotective treatments for POAG. As to translation to individual patients, it is anticipated that as novel drug targets are identified, such as those with a neuroprotective effect, they would benefit many patients, regardless of the specific set of genetic variants each individual harbors.

We found evidence supporting relationships between complex diseases and POAG using MR. For instance, T2D was associated with increased risk of POAG and as well as with IOP levels, consistent with a previous smaller MR study[24] and a meta-analysis of observational studies[25]. However, causality cannot be directly inferred in the case of T2D, because it is possible that a portion of the glaucoma cases in our study (especially from cohort studies such as UK Biobank (UKB)) were diagnosed at a higher rate due to them having had their eyes checked for diabetic retinopathy. We conducted a reverse-direction MR study (POAG → T2D) to try to assess if overdiagnosis of glaucoma in patients with T2D could be a driver of the observed association between T2D and POAG; the MR result was null, suggesting that collectively the

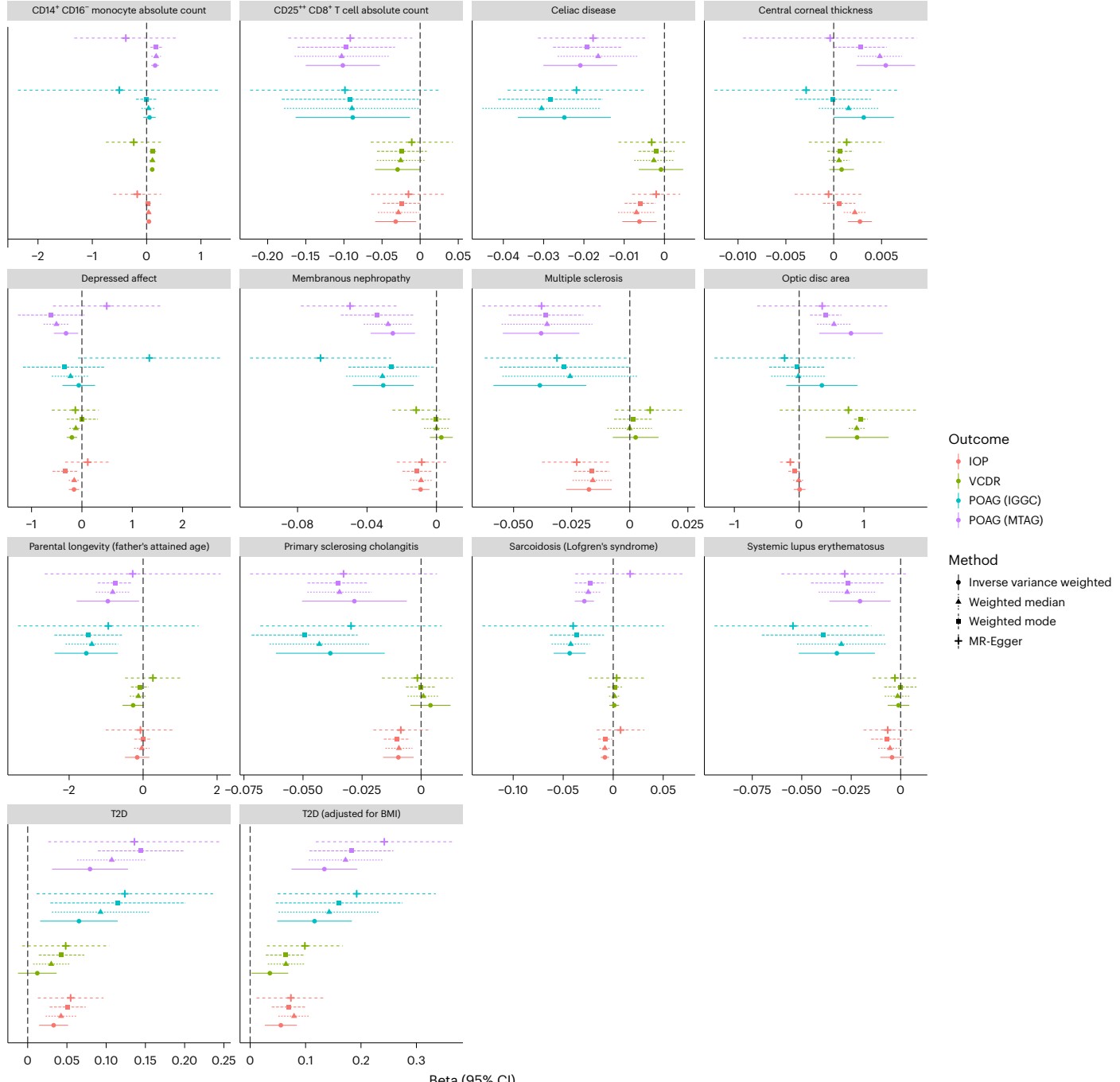

**Fig. 5 | Putative causally associated traits with POAG.** Plots show 14 traits that were associated with POAG from MR (FDR *P* < 0.05, direction: complex traits → POAG). Different outcome traits are shown in different colors. Different MR methods are displayed in different line types. The dots show the effect sizes of MR estimations, and the error bars show the 95% confidence interval of the estimations. The number of SNPs, effect sizes and *P* values are presented in Supplementary Table 14. All tests were two-sided. BMI, body mass index; 95% CI, 95% confidence interval.

diagnosed POAG individuals in our meta-analysis were not strongly enriched for T2D, but interpreting this null MR result is difficult due to limited power given that our MR instrument for POAG is derived from glaucoma samples from heterogeneous sources. Possible biological mechanisms for the effect of diabetes on POAG include compromise of the neuronal and glial functions, and retinal metabolism alterations that cause retinal vascular dysregulation[26].

We also identified several immune disorders that were associated with POAG, such as multiple sclerosis and systemic lupus erythematosus, suggesting that dysregulation of the immune system plays a potential role in glaucomatous optic nerve degeneration.

Previous observational studies have suggested that POAG was related to neuroendocrine-immune abnormalities[27–29]. In our colocalization analysis, we further identified a shared genomics region near *ATXN2* between systemic lupus erythematosus and POAG. Further studies are warranted to characterize the potential underlying pathogenic roles of inflammation or autonomic dysfunction[30,31].

This study has several strengths and limitations. The main strength is a substantial improvement in the statistical power to detect POAG risk loci. Leveraging the high genetic correlation between POAG, IOP and VCDR, we jointly analyzed large-scale GWAS data from these traits obtained through international collaborations and large biobanks. In

addition, the MTAG approach we used in this study accounts for inclusion of datasets with sample overlap (for example, the Canadian Longitudinal Study on Aging (CLSA) endophenotypes and case–control dataset), minimizing possible inflation in test statistics due to such biases. We also observed no evidence for inflation from the FDR measures obtained from the MTAG analysis. Second, almost all the identified POAG loci were replicated in the 23andMe study, indicating the increased power to identify genuine POAG risk loci based on the large-scale multitrait approach. Third, we incorporated GWAS data across ancestries, which further improved the statistical power of this study and helped identify shared risk variants across ancestries. Finally, we integrated transcriptomic and proteomic data using several post-GWAS approaches, which allowed us to identify potential causal genes and druggable targets.

A limitation of this study is the preponderance of European ancestry samples, which reduces power to detect Asian and African ancestry-specific associations. Second, while we have applied various post-GWAS approaches, there remains work to be done to characterize the mechanisms underlying the large number of newly identified loci. Our gene expression-based work uses both retinal tissue as well as a wider range of nonocular tissues. However, our proteomics-based work is based on plasma. Further proteomic studies using more relevant fluids and tissues to evaluate how these risk loci contribute to the pathogenesis of POAG will further shed light on the etiology of the disease. More detailed functional experiments are warranted to delineate the biological mechanisms of each of the identified loci. Finally, we prioritize a list of potential drug targets for POAG based on genetic evidence, but additional functional experiments and clinical trials are needed to support these findings.

In conclusion, our multitrait POAG GWAS has nearly tripled the number of POAG risk loci, with the majority replicated in an independent cohort. Combining multiomics datasets, we have shown the utility of genetic evidence in identifying candidate drug targets for POAG, especially 'neuro-protection' therapeutic targets. We also identified novel associations of POAG with other complex traits, including numerous immune-related diseases. These findings provide insights into the pathogenesis of POAG and enable new drug development for this common cause of blindness.

## Online content

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

Xikun Han [1,2,37] ✉, Puya Gharahkhani [1,2,3,37], Andrew R. Hamel [4,5], Jue Sheng Ong [1], Miguel E. Rentería [2,3,6], Puja Mehta [4,5], Xianjun Dong [7,8], Francesca Pasutto [9], Christopher Hammond [10], Terri L. Young [11], Pirro Hysi [10], Andrew J. Lotery [12,13], Eric Jorgenson [14], Hélène Choquet [15,16], Michael Hauser [17,18,19,20], Jessica N. Cooke Bailey [21,22], Toru Nakazawa [23,24,25,26], Masato Akiyama [16,27], Yukihiro Shiga [23,28,29], Zachary L. Fuller [30], Xin Wang [30], Alex W. Hewitt [31,32], Jamie E. Craig [33], Louis R. Pasquale [34], David A. Mackey [35], Janey L. Wiggs [4,5], Anthony P. Khawaja [36], Ayellet V. Segrè [4,5], 23andMe Research Team[*,**], International Glaucoma Genetics Consortium & Stuart MacGregor [1,2]

[1]Statistical Genetics Lab, QIMR Berghofer Medical Research Institute, Brisbane, Queensland, Australia. [2]Faculty of Medicine, University of Queensland, Brisbane, Queensland, Australia. [3]School of Biomedical Sciences, Faculty of Health, Queensland University of Technology, Brisbane, Queensland, Australia. [4]Department of Ophthalmology, Massachusetts Eye and Ear, Harvard Medical School, Boston, MA, USA. [5]Broad Institute of Harvard and MIT, Cambridge, MA, USA. [6]Mental Health and Neuroscience Program, QIMR Berghofer Medical Research Institute, Brisbane, Queensland, Australia. [7]Genomics and Bioinformatics Hub, Brigham and Women's Hospital, Boston, MA, USA. [8]Department of Neurology, Brigham and Women's Hospital and Harvard Medical School, Boston, MA, USA. [9]Institute of Human Genetics, Universitätsklinikum Erlangen, Friedrich-Alexander-Universität, Erlangen-Nürnberg, Erlangen, Germany. [10]Twin Research and Genetic Epidemiology, King's College London, London, UK. [11]Department of Ophthalmology and Visual Sciences, University of Wisconsin–Madison, Madison, WI, USA. [12]University Hospital Southampton NHS Foundation Trust, Southampton, UK. [13]Faculty of Medicine, University of Southampton, Southampton, UK. [14]Regeneron Pharmaceuticals, Inc., Tarrytown, NY, USA. [15]Division of Research, Kaiser Permanente Northern California (KPNC), Oakland, CA, USA. [16]Department of Ophthalmology, Graduate School of Medical Sciences, Kyushu University, Fukuoka, Japan. [17]Department of Medicine, Duke University, Durham, NC, USA. [18]Department of Ophthalmology, Duke University, Durham, NC, USA. [19]Singapore Eye Research Institute, Singapore, Singapore. [20]Duke-NUS Medical School, Singapore, Singapore. [21]Department of Population and Quantitative Health Sciences, Case Western Reserve University School of Medicine, Cleveland, OH, USA. [22]Cleveland Institute for Computational Biology, Case Western Reserve University School of Medicine, Cleveland, OH, USA. [23]Department of Ophthalmology, Tohoku University Graduate School of Medicine, Sendai, Japan. [24]Department of Retinal Disease Control, Tohoku University Graduate School of Medicine, Sendai, Japan. [25]Department of Advanced Ophthalmic Medicine, Tohoku University Graduate School of Medicine, Sendai, Japan. [26]Department of Ophthalmic Imaging and Information Analytics, Tohoku University Graduate School of Medicine, Sendai, Japan. [27]Laboratory for Statistical Analysis, RIKEN Center for Integrative Medical Sciences, Yokohama, Japan. [28]Department of Neuroscience, Université de Montréal, Montréal, Quebec, Canada. [29]Neuroscience Division, Centre de Recherche du Centre Hospitalier de l'Université de Montréal, Montréal, Quebec, Canada. [30]23andMe, Inc., Sunnyvale, CA, USA. [31]Menzies Institute for Medical Research, University of Tasmania, Hobart, Tasmania, Australia. [32]Centre for Eye Research Australia, University of Melbourne, Melbourne, Victoria, Australia. [33]Department of Ophthalmology, Flinders Medical Centre, Flinders University, Bedford Park, South Australia, Australia. [34]Department of Ophthalmology, Icahn School of Medicine at Mount Sinai, New York, NY, USA. [35]Centre for Ophthalmology and Visual Science, University of Western Australia, Lions Eye Institute, Perth, Western Australia, Australia. [36]NIHR Biomedical Research Centre, Moorfields Eye Hospital NHS Foundation Trust and UCL Institute of Ophthalmology, London, UK. [37]These authors contributed equally: Xikun Han, Puya Gharahkhani. *A list of authors and their affiliations appears at the end of the paper. **A full list of members appears in the supplementary information. ✉e-mail: hanxikun2017@gmail.com

### 23andMe Research Team

### Xin Wang[30] & Zachary L. Fuller[30]

A full list of members appears in the supplementary information.

### International Glaucoma Genetics Consortium

Xikun Han[1,2,37], Puya Gharahkhani[1,2,3,37], Andrew R. Hamel[4,5], Jue Sheng Ong[1], Christopher Hammond[10], Terri L. Young[11], Pirro Hysi[10], Andrew J. Lotery[12,13], Eric Jorgenson[14], Hélène Choquet[15,16], Michael Hauser[17,18,19,20], Jessica N. Cooke Bailey[21,22], Toru Nakazawa[23,24,25,26], Masato Akiyama[16,27], Yukihiro Shiga[23,28,29], Alex W. Hewitt[31,32], Jamie E. Craig[33], Louis R. Pasquale[34], David A. Mackey[35], Janey L. Wiggs[4,5], Anthony P. Khawaja[36], Ayellet V. Segrè[4,5] & Stuart MacGregor[1,2]

## Methods

This study was approved by the QIMR Berghofer Human Research Ethics Committee. In addition, relevant details for each of the participating cohorts are provided below and in 'Ethics statement' section.

### Study populations

In this study, we included genetic and phenotypic data from UKB, CLSA, Mass General Brigham Biobank and our previously published GWAS for POAG, VCDR and IOP (Fig. 1)[6,8,9,32]. The detailed information for each study is described below.

### UKB

The UKB is a large-scale population-based cohort study with deep phenotypic and genetic data from half a million participants aged 40–69 yr from the United Kingdom[33]. For genetic data, approximately 488,000 participants were genotyped with more than 800,000 markers. The genotype platforms, genetic quality controls and imputation procedures were detailed in a previous study[33]. In the current analysis, we included 438,870 participants who were genetically defined as 'white-British' ancestry[8,32]. SNPs with minor allele frequency (MAF) > 0.01 and imputation quality score > 0.8 were kept in association analysis.

The detailed definitions of phenotypic data, including glaucoma, VCDR and IOP, were described in our previous studies[8,9,32,34]. Briefly, glaucoma cases were ascertained from ICD-10 diagnosis, record-linkage data from local general practitioners and self-reported previous diagnosis; controls were defined as participants who self-reported having no eye disease (UKB phenotypic data downloaded in March 2020). In our association analysis, we kept 11,239 glaucoma cases and 137,621 controls of European ancestry. We ran generalized mixed models in SAIGE (v.0.29.6)[35] and adjusted for age, sex and the first ten genetic principal components. In the SAIGE analysis, generalized linear mixed models with two steps were fitted to account for unbalanced case–control ratios and sample relatedness. The first step (fitNULLGLMM) was used to estimate variance component and model parameters. The second step (SPAGMMATtest) performed single variant score tests with saddlepoint approximation based on logistic mixed models[35].

The VCDR measurements of optical nerve head photographs were based on CNN models trained on clinical assessments[9]. Both VCDR and vertical disc diameter from approximately 70,000 UKB fundus images were used to train CNN models. In our previous work, we have shown that AI-based measurements were more accurate and increased GWAS power of genetic discovery. In the current study, we performed GWASs in 68,240 participants with AI labeling VCDR. The association tests were conducted using linear mixed models (BOLT-LMM v.2.3 (ref. 36)) adjusting for vertical disc diameter, age, sex and the first ten principal components.

The IOP GWAS in UKB was based on corneal-compensated IOP measurements in 103,914 participants[8,32]. Linear mixed models were performed in BOLT-LMM (v.2.3) adjusting for age, sex and the first ten principal components.

### CLSA

The CLSA is a cohort study of 51,338 participants aged 45–85 yr from Canada[37,38]. The genetic data (CLSA Baseline Genome-wide Genetic Data Release v.2.0) were available for 19,669 participants genotyped on the Affymetrix Axiom array. The detailed descriptions of genetic quality controls and imputation procedures were presented in the CLSA support document (see 'Data availability' section). In this study, we only included participants of European ancestry based on genetic principal components[9]. SNPs with MAF > 0.01 and imputation quality score > 0.8 were kept in association tests.

For glaucoma status, participants were interviewed in-person at data collection sites, and those who reported yes to the question 'Has a doctor ever told you that you have glaucoma?' were defined as

cases[34]. The remaining participants were defined as controls. In the variant association tests, we included 1,358 glaucoma cases and 16,455 controls using Firth logistic regression in REGENIE (v.1.0.6.2)[39] and we adjusted for age, sex and the first ten genetic principal components.

In the CLSA, retinal fundus images were available for 29,635 participants (106,330 images in total) using a Topcon nonmydriatic retinal camera. The optic nerve head parameters were assessed using the AI models trained in UKB[9]. We included 18,304 participants with both AI labeling VCDR and genetic data. The association tests were conducted by linear mixed models in BOLT-LMM (v.2.3)[36], adjusting for vertical disc diameter, sex, age and the first ten genetic principal components.

The IOP measurements (corneal-compensated IOP) in the CLSA were available for both baseline and follow-up visits on both eyes. We removed measurements <5 mm Hg or >60 mm Hg. Values were averaged across multiple measurements. In total, 18,421 participants were retained in the linear mixed models for association tests (BOLT-LMM v.2.3) adjusting for sex, age and the first ten genetic principal components.

### Mass General Brigham Biobank

Mass General Brigham Biobank (formally known as Partners HealthCare Biobank) is a biorepository of samples from consented patients at Mass General Brigham[40] (parent organization of Massachusetts General Hospital and Brigham and Women's Hospital). In this study, cases were defined based on a diagnosis available on electronic health records, and controls were participants without a recorded diagnosis of glaucoma in their electronic health records. In total, 1,415 glaucoma cases and 18,632 controls were genotyped on an Illumina Multi-Ethnic Global Array (MEGA) (Illumina). Participants showing high rates of missingness or those deemed ancestry outliers from the European ancestry population were removed. Genetic variants with high missingness or extreme allele frequencies were removed before imputation using the HRCr1.1 reference panel (Michigan Imputation server)[41]. Imputed genotype data in dosage format were used for the analysis. Glaucoma GWAS was conducted using PLINK v.2.00 with a logistic regression model adjusting for age, sex, genetic principal components and genotype batches[42].

### The IGGC

The IGGC is a large international consortium established to identify glaucoma genetic risk genes through large-scale meta-analysis[43,44]. For POAG, we previously published a large-scale meta-analysis on 16,677 cases and 199,580 controls of European descent (stage 1 (ref. 6)). In the current study, we included POAG GWAS data on 15,229 POAG cases and 177,473 controls of European descent after excluding UKB samples in our MTAG analysis (described below to optimize the GWAS power and to account for imperfect genetic correlation). We also included GWAS results for VCDR (n = 25,180) and IOP (n = 31,269) of European descent[43,44].

### POAG GWASs of Asian and African ancestry populations from IGGC

In the multiancestry analysis, we included POAG GWAS results from the Asian ancestry population (IGGC, 6,935 cases and 39,588 controls) and the African ancestry population (IGGC, 3,281 cases and 2,791 controls)[6].

### 23andMe replication

The glaucoma cases in the 23andMe study were defined as those who reported glaucoma, excluding angle-closure glaucoma or other types of glaucoma. Participants without glaucoma were defined as controls. In total, 84,910 cases and 2,736,075 controls were included in the GWAS analysis after removing close relatives. In the association tests, logistic regressions were performed in additive models adjusting for age, sex, the first five genetic principal components and genotype platform version. Only the first five principal components were included as a previous study has shown that the first five principal components in the

23andMe dataset explained more variance than the first ten principal components in the UKB, and the total variance in 23andMe reached a plateau after the fifth principal component while in the UKB the variance was flat after the tenth principal component[45]. Participants provided informed consent and participated in the research online, under a protocol approved by the external AAHRPP-accredited IRB, Ethical & Independent Review Services (E&I Review). Participants were included in the analysis on the basis of consent status as checked at the time data analyses were initiated.

## MTAG analysis
MTAG is a generalized meta-analysis method to account for sample overlap, imperfect genetic correlation and genetic heterogeneity across different data sources for the same trait or different traits with a high genetic correlation[7]. In this study, we used a two-stage MTAG approach to meta-analyze POAG, VCDR and IOP data. In the first stage, input datasets for POAG, VCDR and IOP were included in the MTAG analysis separately (three MTAG analyses for POAG and the two endo-phenotypes, respectively).

In the first stage, for POAG MTAG analysis, we included datasets from: (1) 15,229 POAG cases and 177,473 controls of European descent excluding UKB samples; (2) 11,239 glaucoma cases and 137,621 controls of European descent in the UKB; (3) 1,358 glaucoma cases and 16,455 controls of European descent in the CLSA; (4) Mass General Brigham Biobank with 1,415 glaucoma cases and 18,632 controls.

Similarly, for VCDR, we ran MTAG analysis using data from: (1) 68,240 participants with VCDR (adjusted for vertical disc diameter) in the UKB of European descent; (2) 18,304 participants with VCDR (adjusted for vertical disc diameter) in the CLSA of European descent; (3) 25,180 participants with VCDR from IGGC of European descent.

For IOP, we conducted MTAG analysis using data from: (1) 103,914 participants in the UKB of European descent; (2) 18,421 participants in the CLSA of European descent; (3) 31,269 participants from IGGC of European descent.

In the second stage, the trait-specific MTAG outputs from the first stage were further included in MTAG analysis. One key advantage of this two-stage MTAG design was reduced computational burden compared with running MTAG analysis including all GWAS summary statistics for POAG, VCDR and IOP in a single job. In addition, the trait-specific MTAG outputs from the first stage also allowed us to evaluate VCDR- and IOP-specific genetic effects. In our analysis, after filtering out SNPs with MAF < 0.01, 7,259,040 SNPs were kept in the final MTAG output.

## GWAS and cross-ancestry meta-analysis
The association tests in each cohort for various outcomes were described in 'Study populations' section (above). The multiancestry meta-analysis was performed using the fixed-effect inverse variance-weighted method (METAL software 2011-03-25 release[46]) combining POAG MTAG output of European ancestry and POAG GWASs of Asian and African ancestries.

## Definition of independent loci and novel POAG loci
Independent loci were selected using the PLINK clumping method with $P$ value threshold at $5 \times 10^{-8}$, clump $r^2$ 0.01 and a window of 1 Mb for the index variant. 'Novel' POAG loci were defined as independent loci that were not identified in our previous cross-ancestry POAG GWAS[6] or MTAG GWAS[8] (not within ±500 kb of previously reported lead SNPs).

## Proportion of familial risk explained
The proportion of familial risk explained was computed as the sum of $p \times (1 - p) \times b^2/\log_e(9.2)$ over all independent genome-wide significant SNPs (as defined in 'Definition of independent loci and novel POAG loci'), where $p$ is the allele frequency, $b$ is the log odds ratio and 9.2 is the increased risk in first-degree relatives, as estimated in a previous study[47,48].

## Rare variant association analysis from exome sequencing
We compared the common POAG SNPs identified in our MTAG approach with rare variant association results from exome sequencing based on 454,787 UKB participants[12]. The exome sequencing single variant and gene-based association results for glaucoma were obtained from the GWAS Catalog (GCST90079909 and GCST90077754). Significant rare variants or genes were defined as having $P$ values that passed FDR < 5% based on the Benjamini–Hochberg method[49].

## Gene-based and pathway analyses
We conducted gene-based and pathway analyses in MAGMA (v.1.08) as implemented in FUMA (v.1.3.6a)[50,51]. In the gene-based analysis, the association $P$ values of SNPs from GWAS summary statistics were mapped to 18,685 genes, and the derived gene-based $P$ values were adjusted using the Bonferroni method to account for multiple testing ($P < 0.05/18,685 = 2.68 \times 10^{-6}$). In the pathway analysis, the gene-based results were mapped to 15,484 curated gene-sets, and the $P$ values of pathway analysis were adjusted using the Bonferroni method ($P < 0.05/15,484 = 3.23 \times 10^{-6}$).

## Assigning POAG loci into VCDR- or IOP-specific effects
As two key endophenotypes, VCDR and IOP are likely to play distinct roles in the pathological mechanisms of POAG. To investigate the putative role of the identified POAG loci, we applied a hierarchical clustering method to the genetic effects of POAG loci in VCDR and IOP (effect sizes and $z$ scores). Based on the $z$ scores from VCDR and IOP GWASs, the clusters were defined as VCDR- and IOP-specific SNPs. We also performed a multitrait colocalization analysis (HyPrColoc method) to assign POAG loci into VCDR- or IOP-specific effects[52]. In the multitrait colocalization analysis, loci with a high posterior probability supporting a colocalization of POAG and VCDR were defined as VCDR-specific SNPs. Similarly, loci with high posterior probability for POAG and IOP were defined as IOP-specific SNPs.

## Colocalization analysis with eQTL and sQTL data
To prioritize causal genes for the MTAG POAG GWAS loci, we applied the Bayesian-based colocalization method eCAVIAR[53] to each GWAS locus and all overlapping $cis$-eQTLs and $cis$-sQTLs from 49 GTEx tissues (v8)[54] and $cis$-eQTLs from peripheral retina[55]. A colocalization posterior probability above 0.01 was considered significant based on simulations[53]. To minimize false positive results, we removed colocalizing GWAS locus–eQTL/sQTL–gene–tissue results where the eQTL/sQTL and GWAS signals did not pass the following significance cutoffs: eQTL/sQTL FDR < 0.05 or $P < 10^{-4}$ or GWAS $P < 10^{-5}$. Supplementary Tables 7 and 8 present the summary of the colocalization results for the MTAG POAG GWAS loci tested from the European subset and multiancestry meta-analyses, respectively. The colocalizing statistics reported in Results section are only for the GWAS loci that replicated in 23andMe using Bonferroni correction.

## TWAS
We performed TWAS analysis using FUSION software[56] to prioritize potential causal genes using gene expression data from the Eye Genotype Expression database[55] and MTAG POAG GWAS summary statistics. In the TWAS approach, SNPs and gene expression data were used to train gene expression predictive models, which were then used to impute gene–trait association using large-scale GWAS summary statistics based on reference data[56].

## PWAS
PWAS is an approach to evaluate the associations of genetically predicted protein levels and disease outcomes. In PWAS, SNPs and protein expression data were used to train protein expression predictive models, which were further used to impute protein–trait associations using GWAS summary statistics[57]. In the current study, we obtained

predictive protein weight files from 7,213 samples and 1,992 plasma proteins in European Americans[57].

## MR analysis

MR analysis was used to identify plasma proteome or complex diseases or traits that were associated with POAG. We performed two-sample MR analysis to leverage GWAS summary statistics for exposures and outcomes that were derived from different studies or datasets. For exposures with only one SNP as the genetic instrument (that is, plasma proteins), the Wald ratio method was used in the MR analysis. The inverse variance-weighted (MR-IVW) method was used when at least two SNPs were available to perform a weighted linear regression model[58]. We also conducted a series of sensitivity MR analyses to assess the robustness of MR findings that allow violations of MR assumptions, including weighted mode, weighted median and MR-Egger[59–61]. Typically, at least three SNPs were required for these sensitivity MR methods. We used the intercept term from the MR-Egger regression to evaluate directional horizontal pleiotropy effects (that is, intercept close to zero and $P > 0.05$)[60]. The MR pleiotropy residual sum and outlier (MR-PRESSO) method was performed to identify outlier SNPs (MR-PRESSO outlier test) and assess the overall heterogeneity of the MR estimates (MR-PRESSO distortion test)[61].

## MR mapping the effects of plasma proteome on risk of POAG

To facilitate the discovery of drug targets for POAG, we integrated plasma proteome and POAG genetic data in an MR framework. Briefly, 28,191 plasma pQTLs covering 4,719 plasma proteins in 35,559 Icelanders[62] were used as genetic instruments to evaluate putative causal association between plasma proteins and risk of POAG. We applied several different MR methods with different assumptions and advantages, including Wald ratio method, MR-IVW method, weighted median, weighted mode and MR-Egger.

## Drug target prioritization

We used Open Targets[63] to identify drug target genes and investigate the relevance of the corresponding drugs based on their current approved use or clinical trials for other related diseases such as neurodegenerative retinal diseases. We prioritized druggable genes with at least two of the following sources of genetic evidence: (1) proximity to the most significant variants, or genes significant in the MAGMA analysis; (2) genes with eQTL evidence in retina (that is, those that were significant in the TWAS analysis); (3) eQTL/sQTL colocalization in 49 GTEx tissues and retina, investigated using the approach implemented in eCAVIAR; and (4) genes with pQTL evidence in plasma (based on the MR framework for the plasma proteome described above). In addition, to identify drugs with potential neuroprotective effects, we also prioritized drugs for VCDR-specific genes (that is, those associated with VCDR but not IOP) based on the multitrait colocalization and plasma proteome MR analyses for IOP and VCDR, as described earlier.

## Phenome-wide approach identifies genetic correlated traits with POAG

We obtained 1,347 publicly available GWAS summary statistics[64] to systematically evaluate genetically correlated traits with POAG, VCDR and IOP. Linkage disequilibrium score regression was first used to estimate bivariate genetic correlation[65]. We then performed two-sample MR analysis to identify putative causally associated traits with POAG, VCDR and IOP. The MR methods were described in 'MR analysis' section (above). From MR analysis, associated traits with FDR < 0.05 were prioritized to account for multiple testing. For the identified traits, we further performed reverse-directional MR analysis to evaluate the effects of POAG, VCDR or IOP on the associated traits. This analysis can identify 'bidirectional causality' which may reflect a common pathway affecting both the exposure and the outcome[66].

## Shared genomic regions complex traits/diseases and POAG

For the associated complex traits/diseases from the MR analysis, we further conducted a Bayesian colocalization analysis using COLOC (v.5.1.0) to evaluate the shared causal genomic regions[67,68]. The Sum of Single Effects (SuSiE) regression-based colocalization (COLOC-SuSiE) was used where possible to account for multiple causal variants in a region, but falling back on COLOC-single when SuSiE cannot identify any credible sets[68]. The EUR samples in the 1000 Genomes phase 3 data were used to calculate the linkage disequilibrium reference panel[68]. In the colocalization analysis, a posterior probability of H4 (association with both traits) with PP4 > 0.6 was used to support a shared causal variant[67].

## Ethics statement

This study was approved by the QIMR Berghofer Human Research Ethics Committee. In addition, relevant details for each of the participating cohorts are provided below: UK Biobank: The UK Biobank study was approved by the National Health Service National Research Ethics Service (ref. 11/NW/0382) and all participants provided written, informed consent to participate in the UK Biobank study. Information about ethics oversight in the UK Biobank can be found at https://www.ukbiobank.ac.uk/ethics/. CLSA: The CLSA abides by the requirements of the Canadian Institutes of Health Research. The protocol of the CLSA has been reviewed and approved by 13 research ethics boards across Canada. A complete and detailed list is available at: https://www.clsa-elcv.ca/participants/privacy/who-ensures-high-ethical-standards/research-ethics-boards. FinnGen: The Ethical Review Board of the Hospital District of Helsinki and Uusimaa approved the FinnGen study protocol (HUS/990/2017). Mass General Brigham Biobank: Participants in the Mass General Brigham Biobank provided informed consent at sign up and ethics approval was obtained from the Human Research Committee of Mass General Brigham. 23andMe Inc: Participants provided informed consent and participated in the research online, under a protocol approved by the external AAHRPP-accredited IRB, Ethical & Independent Review Services (E&I Review).

## Reporting summary

Further information on research design is available in the Nature Portfolio Reporting Summary linked to this article.

## Data availability

UK Biobank data are available through the UK Biobank Access Management System at https://www.ukbiobank.ac.uk/. Data are available from the Canadian Longitudinal Study on Aging (www.clsa-elcv.ca) for researchers who meet the criteria for access to de-identified CLSA data (https://www.clsa-elcv.ca/researchers/data-support-documentation). The GWAS summary statistics from this study are available for research use at https://xikunhan.github.io/site/publication/. Eye Genotype Expression data are available at the Gene Expression Omnibus (GEO) under accession code GSE115828. The variant-level data for the 23andMe replication dataset are fully disclosed in the paper. Individual-level data are not publicly available due to participant confidentiality, and in accordance with the IRB-approved protocol under which the study was conducted.

## Code availability

The following software packages were used for data analyses: BOLT-LMM software (v.2.3): https://data.broadinstitute.org/alkesgroup/BOLT-LMM; eCAVIAR: https://github.com/fhormoz/caviar; LOCUSZOOM: http://locuszoom.sph.umich.edu/; LD score regression software: https://github.com/bulik/ldsc; METAL software (2011-03-25 release): http://csg.sph.umich.edu/abecasis/Metal/; MTAG (v.1.0.8): Multi-Trait Analysis of GWAS https://github.com/omeed-maghzian/mtag; PLINK software (v.2.00): http://www.cog-genomics.org/plink2;

R: https://cran.r-project.org/; REGENIE software (v.1.0.6.2): https://rgcgithub.github.io/regenie/overview/.

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

## Acknowledgements

The opinions expressed in this manuscript are the authors' own and do not reflect the views of the Canadian Longitudinal Study on Aging (CLSA) or any affiliated institution. This project used data from the UK Biobank under application no. 25331. We acknowledge Mass General Brigham Biobank for providing samples, genomic data and health information data. We acknowledge the participants and investigators of the FinnGen study. This research was made possible using the data/biospecimens collected by the CLSA. Funding for the CLSA is provided by the Government of Canada through the Canadian Institutes of Health Research (CIHR) under grant no. LSA 94473 and the Canada Foundation for Innovation, as well as the following provinces: Newfoundland, Nova Scotia, Quebec, Ontario, Manitoba, Alberta and British Columbia. This research has been

conducted using the CLSA dataset (Baseline Comprehensive Dataset version 4.0, Follow-up 1 Comprehensive Dataset version 1.0), under Application Number 190225. The CLSA is led by P. Raina, C. Wolfson and S. Kirkland. S.M. is supported by a research fellowship from the Australian National Health and Medical Research Council (NHMRC). S.M., D.A.M., J.E.C. and A.W.H. acknowledge Program Grant (grant no. 1150144) and Centre of Research Excellence (grant no. 1116360) Funding from the Australian NHMRC. P.G. is supported by an NHMRC Investigator Grant (#1173390). J.L.W. acknowledges support from NIH NEI R01 EY022305 and P30 EY014104. L.R.P. is supported by Research to Prevent Blindness (NYC, USA), the Glaucoma Foundation (NYC, USA) and grant nos. NEI R01 EY 015473 and EY 032559. A.V.S., A.R.H. and P.M. were supported by NIH/NEI grant no. R01 EY031424. A.P.K. is supported by a UK Research and Innovation Future Leaders Fellowship, an Alcon Research Institute Young Investigator Award and a Lister Institute for Preventive Medicine Award. This research was supported by the NIHR Biomedical Research Centre at Moorfields Eye Hospital and the UCL Institute of Ophthalmology. We acknowledge a Core Grant for Vision Research from the National Eye Institute/ National Institutes of Healthto the University of Wisconsin–Madison (P30EY016665) and an Unrestricted Grant from Research to Prevent Blindness,Inc. to the UW-Madison Department of Ophthalmology and Visual Sciences. We acknowledge the donors of National Glaucoma Research, a program of the BrightFocus Foundation, for support of this research (BrightFocus Grant Submission Number: G2021009S).

## Author contributions

S.M., X.H. and P.G. designed the study. X.H., P.G., J.S.O., M.E.R., X.D., A.R.H., P.M., A.V.S., Z.L.F., X.W. and S.M. analyzed the data. S.M., P.G., A.W.H., J.E.C., J.L.W. and D.A.M. obtained funding. S.M., X.H., P.G., M.E.R., X.D., F.P., C.H., T.L.Y., P.H., A.W.H., J.E.C., L.R.P., D.A.M., J.L.W., A.P.K., A.J.L., E.J., H.C., M.H., J.N.C.B., T.N., M.A. and Y.S. contributed to data collection and contributed to genotyping. X.H., P.G. and S.M. wrote the first draft of the paper. All authors contributed to the final version of the paper.

## Competing interests

L.R.P. is a consultant to Eyenovia, Twenty Twenty and Skye Bioscience. S.M., J.E.C. and A.W.H. are co-founders of and hold stock in Seonix Pty Ltd. Z.L.F. and X.W. are employed by and hold stock or stock options in 23andMe, Inc. A.P.K. has acted as a paid consultant or lecturer to Abbvie, Aerie, Allergan, Google Health, Heidelberg Engineering, Novartis, Reichert, Santen and Thea. The other authors declare no competing interests.

## Additional information

**Extended data** is available for this paper at https://doi.org/10.1038/s41588-023-01428-5.

**Correspondence and requests for materials** should be addressed to Xikun Han.

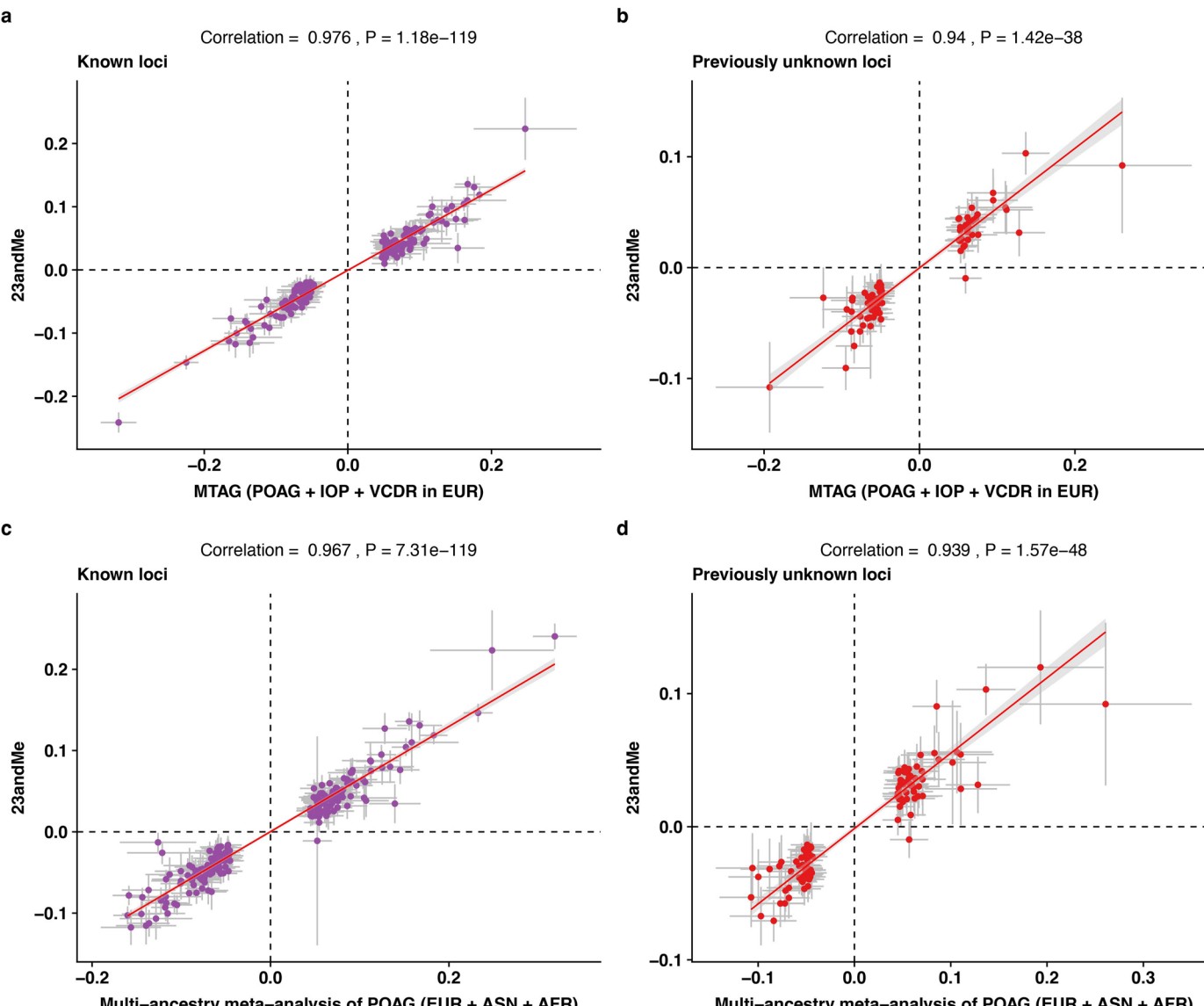

**Extended Data Fig. 1 | Comparison of the effect sizes for genome-wide significant independent SNPs by known loci and previously unknown loci.** **a**, Plot showing effect sizes for known genome-wide significant independent SNPs identified from the POAG multi-trait GWAS in European ancestry versus glaucoma GWAS in 23andMe. **b**, Plot showing 81 previously unknown genome-wide significant independent SNPs identified from the POAG multi-trait GWAS in European ancestry versus glaucoma GWAS in 23andMe. The Pearson's coefficient is 0.94 ($P = 1.42 \times 10^{-38}$). **c**, Plot showing effect sizes for known genome-wide significant independent SNPs identified from the POAG multi-ancestry meta-analysis versus glaucoma GWAS in 23andMe. **d**, Plot showing 109 previously unknown genome-wide significant independent SNPs identified from POAG multi-ancestry meta-analysis versus glaucoma GWAS in 23andMe. The Pearson's coefficient is 0.939 ($P = 1.57 \times 10^{-48}$). For the 81 previously unknown genome-

wide significant independent SNPs identified from the POAG multi-trait GWAS in European ancestry, the replication rates in an independent cohort using 23andMe were: 38% SNPs ($n = 31$) passed the genome-wide significance level ($P < 5 \times 10^{-8}$) in the 23andMe study, 73% SNPs ($n = 59$) were significant after Bonferroni correction ($P < 0.00062$), and 96% SNPs ($n = 78$) reached a nominal significance level ($P < 0.05$). For the 109 previously unknown genome-wide significant independent SNPs identified from POAG multi-ancestry meta-analysis, the replication rates in an independent cohort using 23andMe were: 38% SNPs ($n = 39$) passed the genome-wide significance level ($P < 5 \times 10^{-8}$) in the 23andMe study, 66% SNPs ($n = 68$) were significant after Bonferroni correction ($P < 0.0005$), and 96% SNPs ($n = 99$) reached a nominal significance level ($P < 0.05$). The dots show the effect sizes of SNPs, and the error bars show the 95% confidence interval of the estimations of SNP effect sizes.

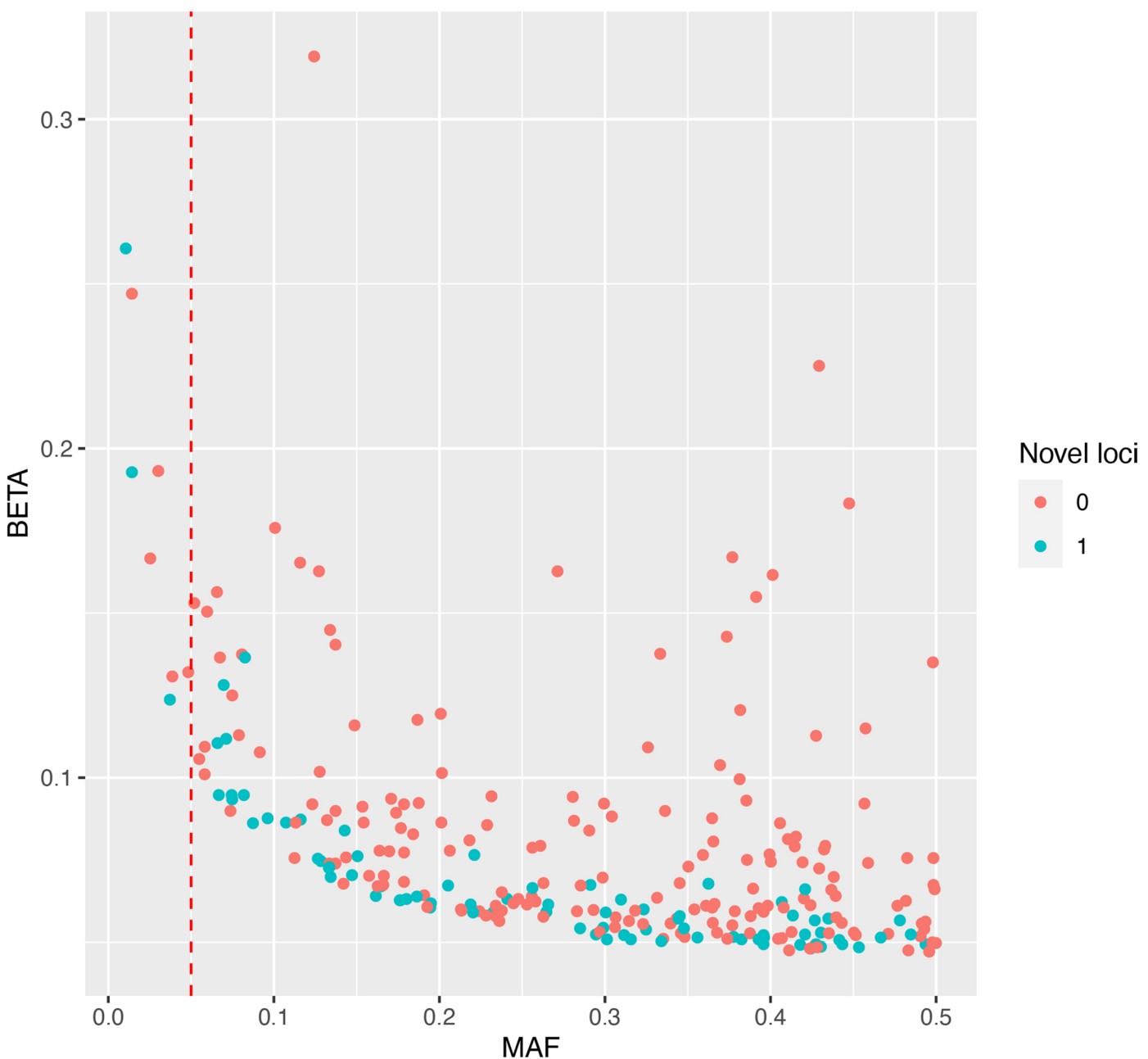

**Extended Data Fig. 2 | Scatterplot of minor allele frequency (MAF) and effect size (absolute value) for the novel loci in the multi-trait POAG GWAS analysis.** '1' corresponds to novel SNPs; '0' corresponds to known SNPs.

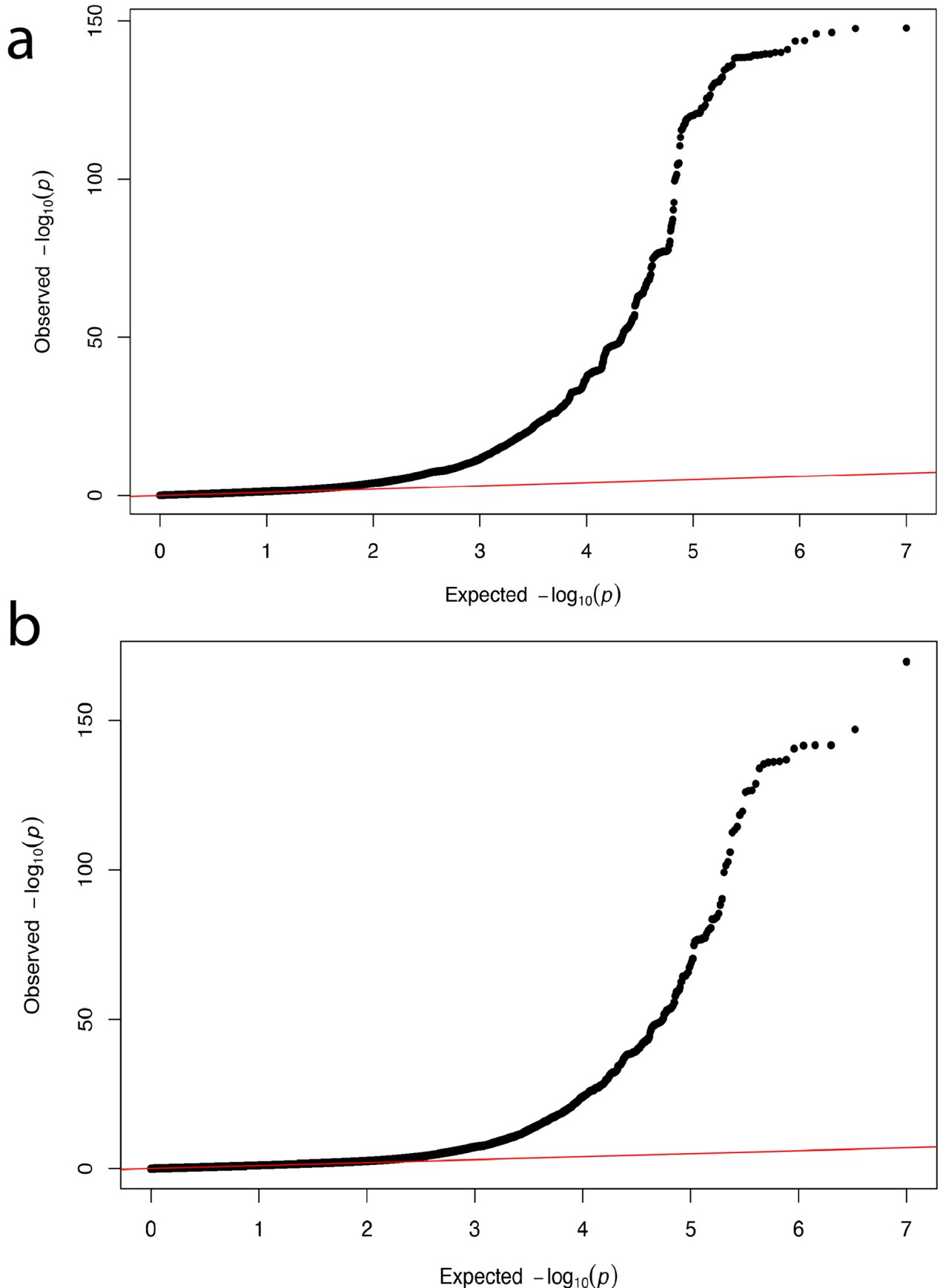

**Extended Data Fig. 3 | Quantile-quantile plots for POAG GWAS. a**, Quantile-quantile plot for multi-trait POAG GWAS in participants of European ancestry. The quantile-quantile plot is based on one million randomly selected SNPs. Linkage disequilibrium (LD) score regression intercept is used to assess the genomic inflation; the intercept is 0.957 (standard error = 0.013, attenuation ratio < 0), and the lambda value is 1.27. **b**, Quantile-quantile plot for POAG multi-ancestry meta-analysis. The lambda value is 1.28. Because of the multi-ancestry design, LDSC was not performed.

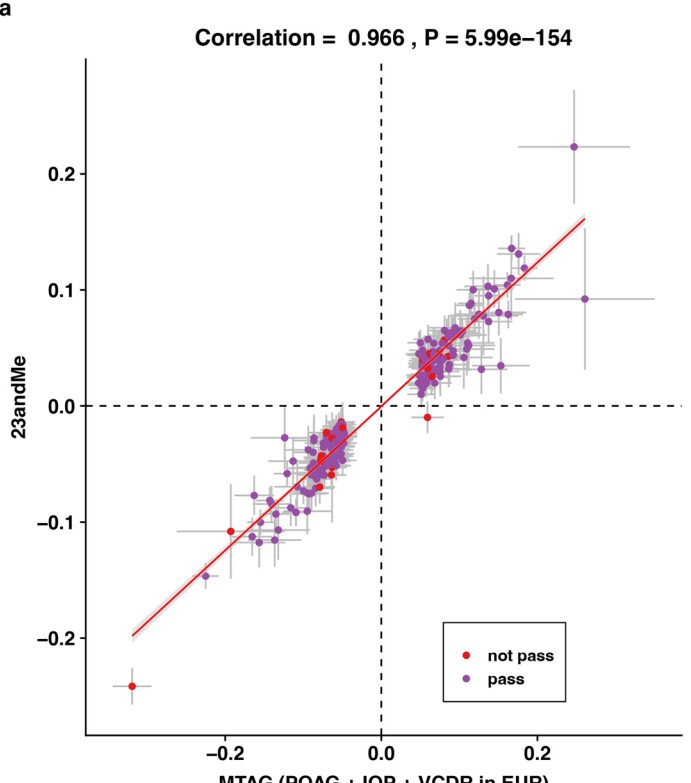

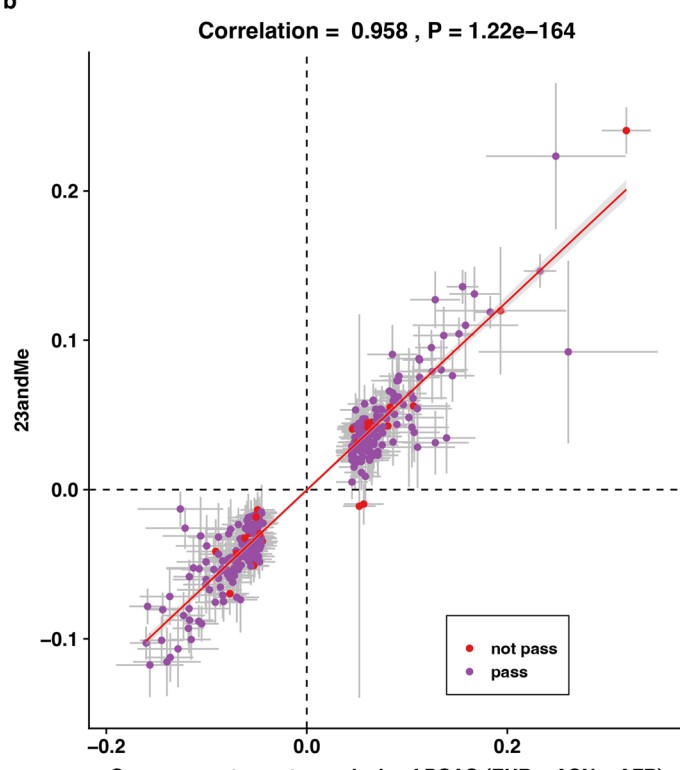

**Extended Data Fig. 4 | Replication in 23andMe based on quality control annotation.** Comparison of the effect sizes (log odds ratio) for genome-wide significant independent SNPs in our discovery studies and the 23andMe replication cohort. **a**, Plot shows genome-wide significant independent SNPs identified from the POAG multi-trait GWAS in the European population versus glaucoma GWAS in 23andMe (*n* = 261 independent SNPs). **b**, Plot shows genome-wide significant independent SNPs identified from the POAG multi-ancestry meta-analysis versus glaucoma GWAS in 23andMe (*n* = 302 independent SNPs). The two different colors for 'not pass' and 'pass' represent the quality control annotation from 23andMe. Most of the 27 SNPs that did not pass the quality control annotated by 23andMe showed high concordance in effect size between the discovery and replication cohorts. The dots show the effect sizes of SNPs, and the error bars show the 95% confidence interval of the estimations of SNPs effect sizes.

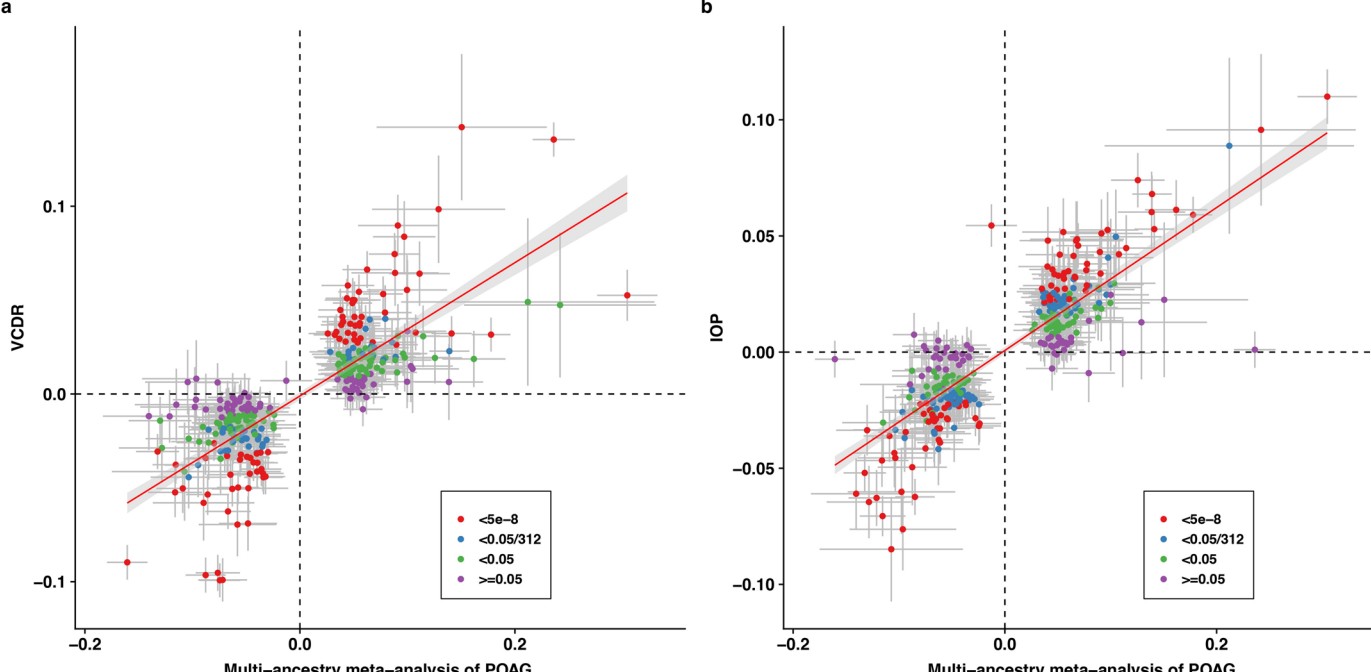

**Extended Data Fig. 5 | Comparison of the effect sizes for 312 POAG genome-wide significant independent SNPs from multi-ancestry meta-analysis against their effect sizes in VCDR and IOP grouped by different P-value thresholds. a**, Plot showing the comparison with VCDR (*n* = 312 independent SNPs). The *x*-axis shows the effect sizes in multi-ancestry meta-analysis of POAG. The *y*-axis shows the effect sizes in VCDR. The SNPs are shown in different colors based on different *P* values in VCDR ($P < 5 \times 10^{-8}$; '<0.05/312': $5 \times 10^{-8} \leq P < 0.05/312$; '<0.05': $0.05/312 \leq P < 0.05; P \geq 0.05$). **b**, Plot showing the comparison with IOP (*n* = 312 independent SNPs). The SNPs are shown in

different colors based on different *P* values in IOP ($P < 5 \times 10^{-8}$; '<0.05/312': $P < 5 \times 10^{-8} \leq P < 0.05/312$; '<0.05': $0.05/312 \leq P < 0.05; P \geq 0.05$). The dots show the effect sizes of SNPs, and the error bars show the 95% confidence interval of the estimations of SNPs effect sizes. In our previous work, we have shown that the genetic correlation between VCDR and POAG is 0.5, and many VCDR significant loci are not necessarily associated with POAG. In this figure, many POAG SNPs are not associated with VCDR, and similarly many VCDR SNPs are not associated with POAG.

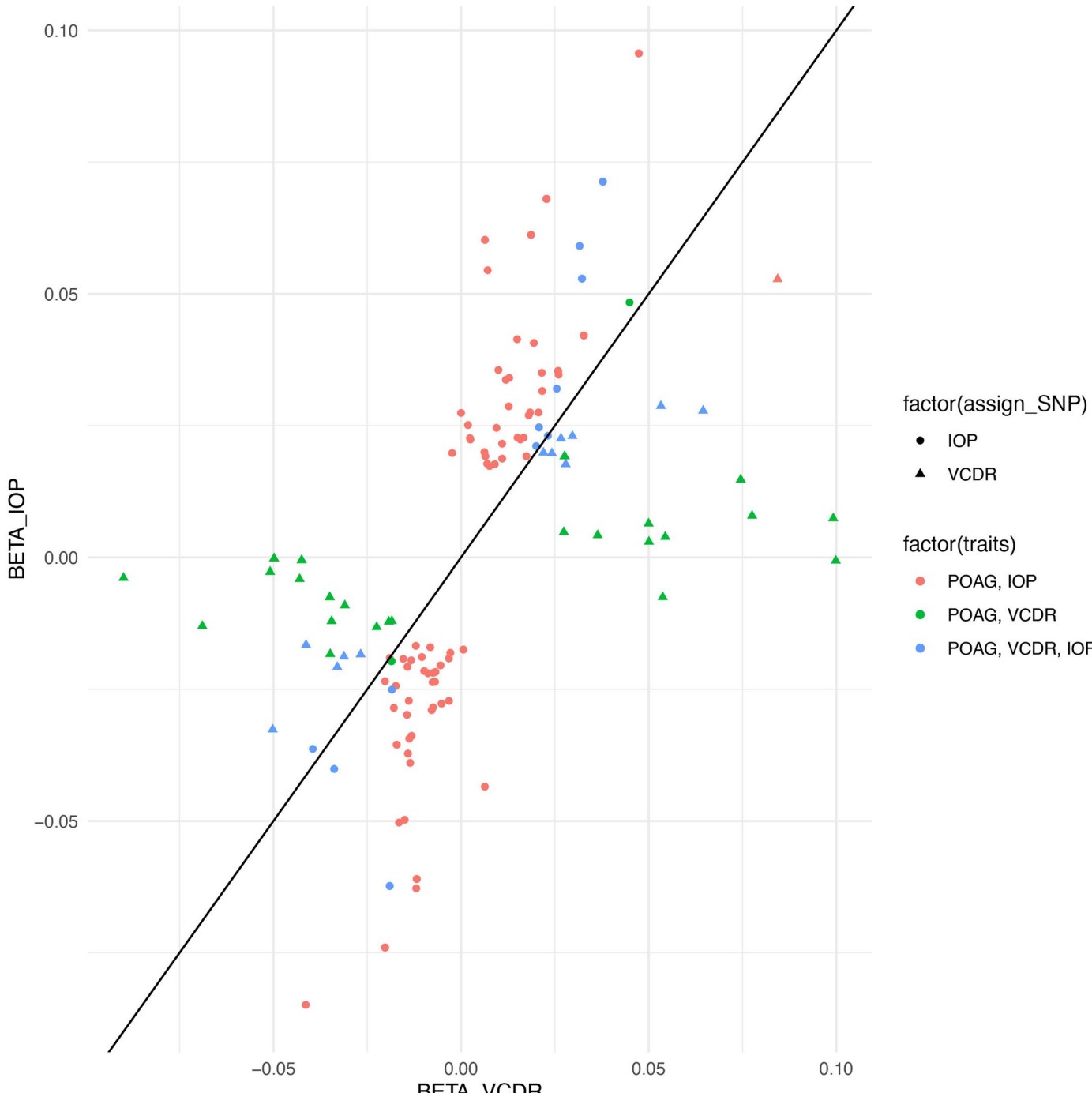

**Extended Data Fig. 6 | Classification of POAG loci into VCDR or IOP-specific SNPs based on multi-trait colocalization.** This scatter plot shows the effect sizes of 263 MTAG POAG loci on VCDR (*x*-axis) and IOP (*y*-axis). The assigned SNPs (IOP and VCDR in different point shapes) were based on a hierarchical clustering approach. The different point colors for trait combination show the results from multi-trait colocalization. This figure shows a consistent classification of POAG loci into VCDR- and IOP-specific SNPs based on the hierarchical clustering method and the multi-trait colocalization method.

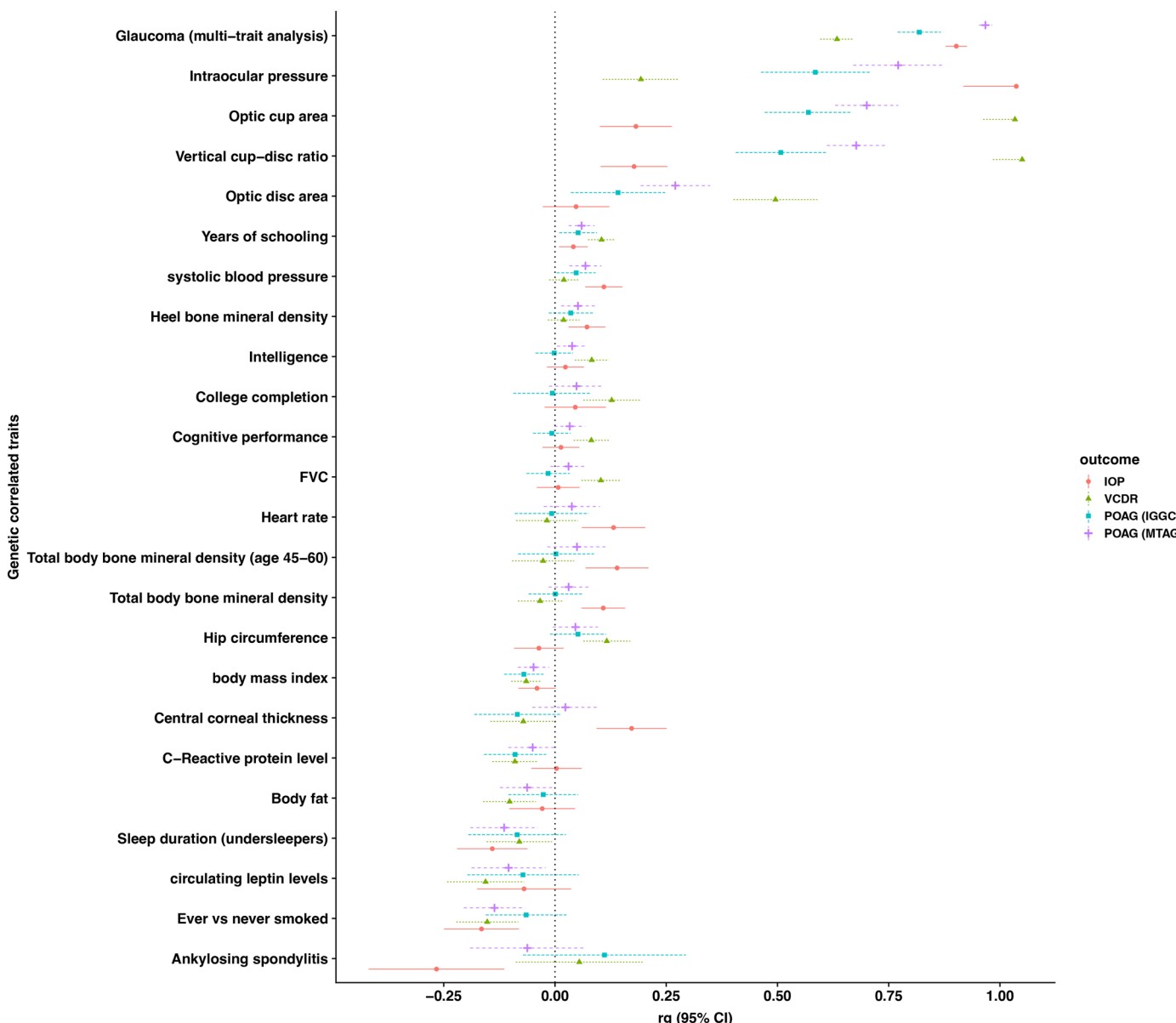

**Extended Data Fig. 7 | Bivariate genetic correlation analysis identifies 24 traits that are genetically correlated with POAG, VCDR or IOP.** The x-axis shows the genetic correlations and their 95% confidence intervals. The y-axis shows the genetically correlated traits. The dots show the effect size of Mendelian randomization estimations, and the error bars show the 95% confidence interval of the estimations. All tests were two-sided.

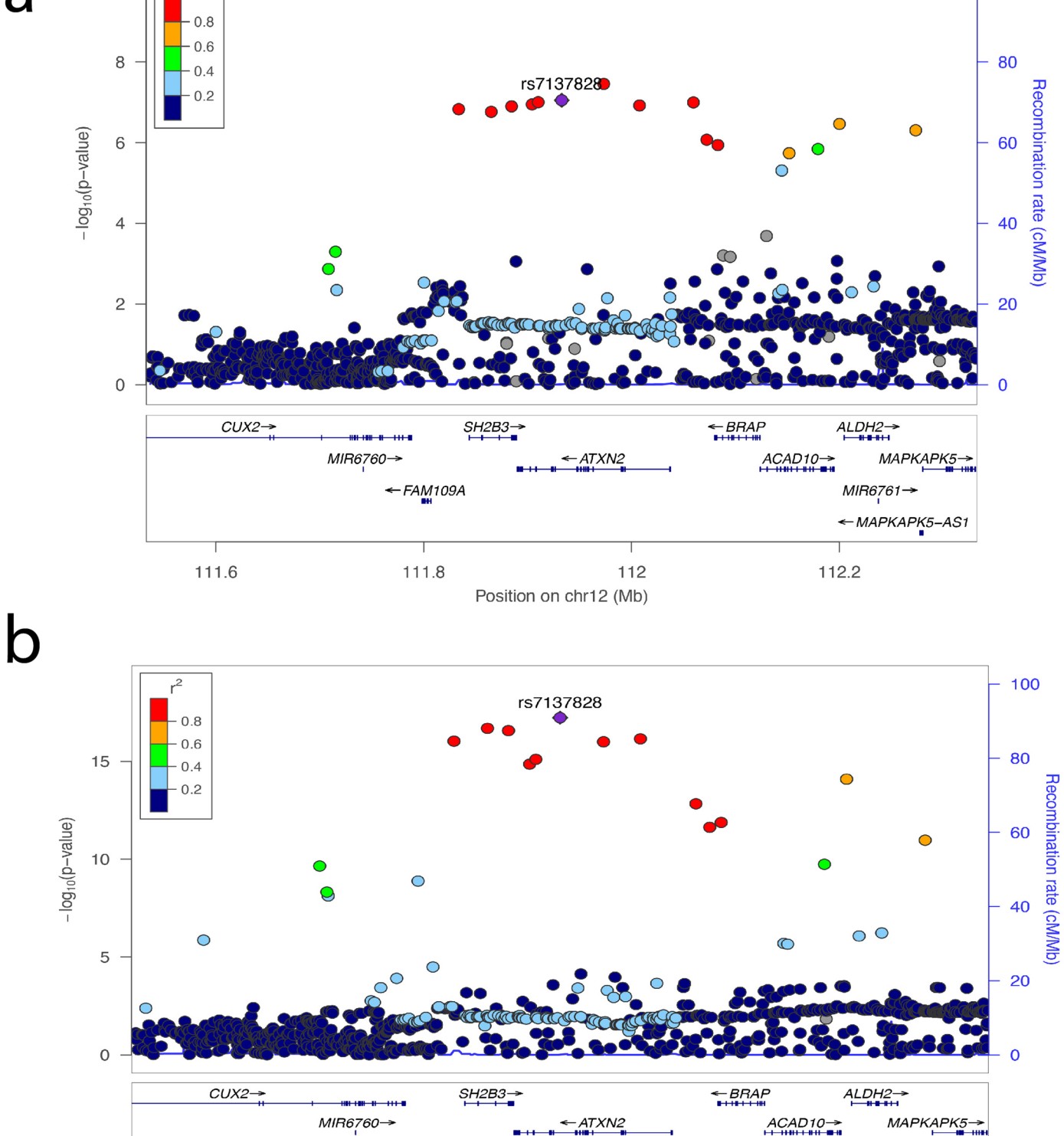

**Extended Data Fig. 8 | Shared genomic region between systemic lupus erythematosus and POAG.** The genomic region (gene *ATXN2*) has a high posterior probability (PP4 = 0.98) in the Bayesian colocalization analysis. **a**, LocusZoom plot for systemic lupus erythematosus. **b**, LocusZoom plot for POAG.

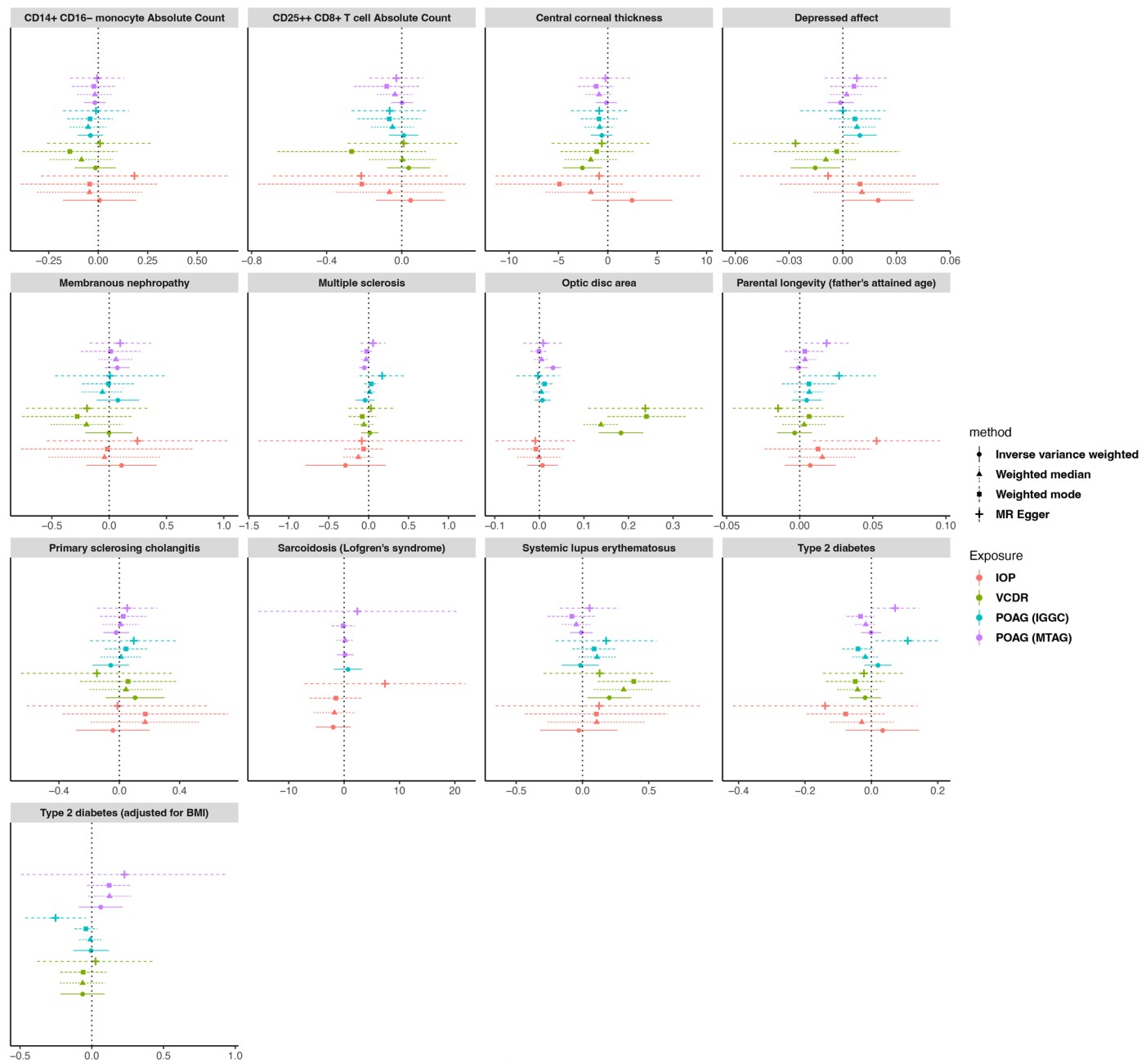

**Extended Data Fig. 9 | Reverse-directional Mendelian randomization analysis for associated traits with POAG.** Plots show the reverse-directional Mendelian randomization for traits that were associated with POAG (celiac disease was not shown because of the limited number of overlapping SNPs; direction: POAG -> complex traits). Different exposure traits are shown in different colors. Different MR methods are displayed in different line types. In this reverse-directional MR analysis, only the association between VCDR and optic disc area passed multiple testing.

# Reporting Summary

## Statistics

For all statistical analyses, confirm that the following items are present in the figure legend, table legend, main text, or Methods section.

| n/a | Confirmed | |
|---|---|---|
| ☐ | ☒ | The exact sample size (*n*) for each experimental group/condition, given as a discrete number and unit of measurement |
| ☐ | ☒ | A statement on whether measurements were taken from distinct samples or whether the same sample was measured repeatedly |
| ☐ | ☒ | The statistical test(s) used AND whether they are one- or two-sided <br> *Only common tests should be described solely by name; describe more complex techniques in the Methods section.* |
| ☐ | ☒ | A description of all covariates tested |
| ☐ | ☒ | A description of any assumptions or corrections, such as tests of normality and adjustment for multiple comparisons |
| ☐ | ☒ | A full description of the statistical parameters including central tendency (e.g. means) or other basic estimates (e.g. regression coefficient) AND variation (e.g. standard deviation) or associated estimates of uncertainty (e.g. confidence intervals) |
| ☐ | ☒ | For null hypothesis testing, the test statistic (e.g. *F*, *t*, *r*) with confidence intervals, effect sizes, degrees of freedom and *P* value noted <br> *Give P values as exact values whenever suitable.* |
| ☒ | ☐ | For Bayesian analysis, information on the choice of priors and Markov chain Monte Carlo settings |
| ☒ | ☐ | For hierarchical and complex designs, identification of the appropriate level for tests and full reporting of outcomes |
| ☐ | ☒ | Estimates of effect sizes (e.g. Cohen's *d*, Pearson's *r*), indicating how they were calculated |

*Our web collection on statistics for biologists contains articles on many of the points above.*

## Software and code

Policy information about availability of computer code

| Data collection | In this study, we included genetic and phenotypic data from UK Biobank (UKB), Canadian Longitudinal Study on Aging (CLSA), Mass General Brigham Biobank, and our previously published GWAS for POAG, VCDR, and IOP. The detailed information for each study is described in the methods section. |
|---|---|
| Data analysis | In this study, we performed various genetic analysis, including genome-wide association study and cross-ancestry meta-analysis, multi-trait GWAS (MTAG) analysis, colocalization analysis, Mendelian randomization analysis, proteome-wide association study, etc. The detailed information is described in the methods section. <br> The following software packages were used for data analyses: BOLT-LMM software (version 2.3): https://data.broadinstitute.org/alkesgroup/BOLT-LMM; eCAVIAR: https://github.com/fhormoz/caviar; LOCUSZOOM: http://locuszoom.sph.umich.edu/ ; LD score regression software: https://github.com/bulik/ldsc; METAL software (2011-03-25 release): http://csg.sph.umich.edu/abecasis/Metal/; MTAG (v1.0.8): Multi-Trait Analysis of GWAS https://github.com/omeed-maghzian/mtag; PLINK software ( v2.00): http://www.cog-genomics.org/plink2; R: https://cran.r-project.org/; REGENIE software (v1.0.6.2): https://rgcgithub.github.io/regenie/overview/. |

For manuscripts utilizing custom algorithms or software that are central to the research but not yet described in published literature, software must be made available to editors and reviewers. We strongly encourage code deposition in a community repository (e.g. GitHub). See the Nature Portfolio guidelines for submitting code & software for further information.

## Data

Policy information about availability of data

All manuscripts must include a data availability statement. This statement should provide the following information, where applicable:

- Accession codes, unique identifiers, or web links for publicly available datasets
- A description of any restrictions on data availability
- For clinical datasets or third party data, please ensure that the statement adheres to our policy

UK Biobank data are available through the UK Biobank Access Management System https://www.ukbiobank.ac.uk/.
Data are available from the Canadian Longitudinal Study on Aging (www.clsa-elcv.ca) for researchers who meet the criteria for access to de-identified CLSA data (https://www.clsa-elcv.ca/researchers/data-support-documentation).
The GWAS summary statistics from this study are available for research use at https://xikunhan.github.io/site/publication/.
Eye Genotype Expression data are are available at Gene Expression Omnibus (GEO) under accession code GSE115828.
The variant-level data for the 23andMe replication dataset are fully disclosed in the manuscript. Individual-level data are not publicly available due to participant confidentiality, and in accordance with the IRB-approved protocol under which the study was conducted.

# Field-specific reporting

Please select the one below that is the best fit for your research. If you are not sure, read the appropriate sections before making your selection.

☒ Life sciences ☐ Behavioural & social sciences ☐ Ecological, evolutionary & environmental sciences

For a reference copy of the document with all sections, see nature.com/documents/nr-reporting-summary-flat.pdf

# Life sciences study design

All studies must disclose on these points even when the disclosure is negative.

| | |
|---|---|
| Sample size | We assembled the largest possible sample size to maximize the number of novel loci. |
| Data exclusions | This is described in the methods. Some samples were excluded based on genetic control procedures and genetic ancestry to ensure homogeneity. |
| Replication | The identified genetic loci were replicated in a large independent cohort from 23andMe, Inc (total sample size>2.8 million). For drug target discovery, we performed genetic analysis from multi-datasets to provide multi-level evidence. All data sets were analyzed independently to replicate genetic loci. |
| Randomization | Samples were from collected study cohorts and were not randomized (randomization is not applicable here). In our genetic association analysis, we adjust for age, sex, and genetic principal components to control covariates. |
| Blinding | Genotyping and quality control for the genetic data was conducted without knowledge of the phenotypes. |

# Reporting for specific materials, systems and methods

We require information from authors about some types of materials, experimental systems and methods used in many studies. Here, indicate whether each material, system or method listed is relevant to your study. If you are not sure if a list item applies to your research, read the appropriate section before selecting a response.

### Materials & experimental systems

| n/a | Involved in the study |
|---|---|
| ☒ | ☐ Antibodies |
| ☒ | ☐ Eukaryotic cell lines |
| ☒ | ☐ Palaeontology and archaeology |
| ☒ | ☐ Animals and other organisms |
| ☐ | ☒ Human research participants |
| ☒ | ☐ Clinical data |
| ☒ | ☐ Dual use research of concern |

### Methods

| n/a | Involved in the study |
|---|---|
| ☒ | ☐ ChIP-seq |
| ☒ | ☐ Flow cytometry |
| ☒ | ☐ MRI-based neuroimaging |

## Human research participants

Policy information about studies involving human research participants

| | |
|---|---|
| Population characteristics | . In this study, we included genetic and phenotypic data from UK Biobank (UKB), Canadian Longitudinal Study on Aging |

| Population characteristics | (CLSA), Mass General Brigham Biobank, and our previously published GWAS for POAG, VCDR, and IOP. Each study was described in full in the methods section. |
| --- | --- |
| Recruitment | Described in full in the methods section. |
| Ethics oversight | Described in full in the methods section. |

Note that full information on the approval of the study protocol must also be provided in the manuscript.

