## [Peer Review File · Nature Genetics]

Peer Review Information

Manuscript Title: Large-scale multi-trait genome-wide association analyses identify hundreds of glaucoma risk loci

Corresponding author name(s): Dr Xikun Han

Reviewer Comments & Decisions:

Decision Letter, initial version:
--

10th June 2022

Dear Xikun,

Your Article "Large scale multi-trait genome-wide association analysis identifies hundreds of glaucoma risk loci" has been seen by two referees. You will see from their comments below that, while they find your work of interest, they have raised several important points. We are interested in the possibility of publishing your study in Nature Genetics, but we would like to consider your response to these points in the form of a revised manuscript before we make a final decision on publication.

To guide the scope of the revisions, the editors discuss the referee reports in detail within the team, including with the chief editor, with a view to identifying key priorities that should be addressed in revision, and sometimes overruling referee requests that are deemed beyond the scope of the current study. In this case, we particularly ask that you revise the presentation to clarify the advances over previous work and the specific criteria used to classify discoveries as novel in the current analysis, and that you address all technical queries related to the association analyses and their interpretation. We hope you will find this prioritized set of referee points to be useful when revising your study. Please do not hesitate to get in touch if you would like to discuss these issues further.

We therefore invite you to revise your manuscript taking into account all reviewer and editor comments. Please highlight all changes in the manuscript text file. At this stage we will need you to upload a copy of the manuscript in MS Word .docx or similar editable format.

*1) Include a "Response to referees" document detailing, point-by-point, how you addressed each

referee comment. If no action was taken to address a point, you must provide a compelling argument. This response will be sent back to the referees along with the revised manuscript.

*2) If you have not done so already please begin to revise your manuscript so that it conforms to our Article format instructions, available [here](http://www.nature.com/ng/authors/article_types/index.html). Refer also to any guidelines provided in this letter.

[redacted]

We hope to receive your revised manuscript within 8-12 weeks. If you cannot send it within this time, please let us know.

Sincerely,
Kyle

Kyle Vogan, PhD

Senior Editor
Nature Genetics
<https://orcid.org/0000-0001-9565-9665>

Referee expertise:

Referee #1: Genetics, vision disorders, bioinformatics

Referee #2: Genetics, statistical methods, complex traits

Reviewers' Comments:

Reviewer #1:
Remarks to the Author:

Han and coauthors conducted large-scale multi-trait analyses and meta-analyses and identified novel genetic loci associated with POAG. They also prioritized possible causal genes and drug targets for POAG by querying a number of omics datasets. The authors further carried out MR analysis. There are several strengths, e.g. the sample size is impressive and the data analysis is extensive. This manuscript reminded me of the Craig et al. 2020 Nat Genet paper (from the same group) that I reviewed previously. This manuscript follows a similar design, i.e. MTAG, but with more samples from CLSA (1,358/16,455), MGB (1,415/18,632) and IGGC (not clear how many more samples than their previous meta-analyses). Despite the novel loci identified and the bioinformatics queries on potential causal genes and drug targets, the translational implications of these loci are still not clear.

1. With more samples included, it is likely that we will identify more loci with weaker effect sizes. However, how do these new loci of much weaker effect sizes contribute to diagnosis and treatment? Do they further improve the prediction of POAG, and by how much?
2. With respect to drug targets, many pharmaceutical companies, including the cited Regeneron's WES paper, are using rare variants with large effect sizes. Common variants typically have smaller effect sizes. Furthermore, among the 312 POAG loci identified, only about 25 (8%) are included in Table 1, prioritized drug targets. What about the rest of the majority of POAG loci? How do these results translate to individual patients, who may have different causes from the 312 POAG loci?
3. The authors also investigated potential causal genes. However, without the support from wet-lab experiments, it is challenging to pin down causal genes/variants.
4. There is some inconsistency in the data analysis. For example, the authors used linear mixed models in their UKBB VCDR analysis (assuming related subjects were included) but required unrelated samples for their UKBB glaucoma analysis. There are methods, e.g. SAIGE, that can handle related subjects in case-control glaucoma analysis. The authors used ten genetic principal components (PCs) in their UKBB analysis but only used 5 PCs in their 23andMe analysis. The 23andMe dataset with millions of samples is likely to be more heterogeneous than the UKBB dataset and ideally more PCs should be used.

5. Comparing to the very recent Gharahkhani et al (2021) meta-analysis with 34,179 cases and 349,321 controls from many of the same authors, what are the reasons that this meta-analysis of less samples of 29,241 cases and 350,181 controls identified many more loci than the previous very similar cross-ancestry meta-analysis and replication in 23andMe?

Reviewer #2:

Remarks to the Author:

Han et al. describe results from multi-trait and multi-ancestry GWAS for glaucoma. Their multi-trait GWAS utilizes a well-established multi-trait framework (MTAG). Compared to single-outcome GWAS, MTAG yields higher power by exploiting the correlation between traits. The authors incorporate three glaucoma outcomes in MTAG: (i) case/control data on primary open-angle glaucoma (POAG; most common form of glaucoma; cases defined via electronic health record or self-report), (ii) intraocular pressure (IOP; known quantitative risk factor for glaucoma; measured via corneal-compensated IOP measurements) and (iii) vertical cup-disc ratio (VCDR, quantifies optic nerve head damage, known marker for glaucoma; assessed via convolutional neural net models trained on optical nerve head photographs). MTAG analyses were based on Europeans-only. In comparison, their multi-ancestry GWAS (adding data from Asian and African populations) were only conducted for POAG: They conducted single-outcome GWAS on POAG separately in Europeans, Asians and Africans and then combined ancestry-specific results via fixed-effect meta-analyses. They did not employ a multi-trait framework for the multi-ancestry analyses. Analyses are based on data from UK Biobank, Canadian Longitudinal Study on Aging, Mass General Brigham Biobank and International Glaucoma Genetic Consortium. This is an appropriate and powerful sample for this kind of study.

Their MTAG analyses based on Europeans identified 263 independent glaucoma-associated loci, of which 81 are claimed to be novel for glaucoma. Their multi-ancestry POAG GWAS identified 312 independent loci, of which 102 are claimed to be novel for glaucoma. Importantly, the authors provide impressive replication of their findings in a large data set from 23andMe with ~80% of identified loci being replicated at a robust Bonferroni-corrected alpha-level.

The authors provide an impressive amount and state-of-the art multi-omics GWAS follow-up analyses to prioritize genes at the identified loci. These include analyses of eQTL and sQTL colocalization (GTEx and retina tissue), transcriptome-wide association (in retina tissue), plasma proteome based on Mendelian Randomization. They identify 69 genes that are known drug targets genes for other diseases and claim them as "drug targets for POAG". Among the 69, the authors highlight 17 genes that were supported by at least two lines of post-GWAS evidence and 10 that were located in VCDR-specific loci. The latter are claimed to be potential targets for neuroprotective treatments that may act through optic nerves, which are unavailable to date for glaucoma (all established treatments are on IOP). This is a relevant result. Finally, the authors found several complex traits/disease genetically correlated with glaucoma by LD score regression and Mendelian randomization analyses. No fine-mapping of causal variant resolution was performed. While the analyses are impressive and the results (especially the neuroprotective drug candidates) are highly relevant for glaucoma drug development, I have the following major concerns:

1. Description of main analyses

The title states "multi-trait GWAS identified hundreds of glaucoma loci". The abstract methods state "we conducted ... multi-trait GWAS". Yet, multi-trait GWAS results are not described in the abstract. Instead, results from a multi-ancestry meta-analysis of POAG (= single trait) are presented in the abstract: The authors write "Employing a multi-ancestry approach, we increased the number of independent risk loci to 312". This is misleading because the 312 were not identified from a multi-trait analysis. I expect to read the main result in the abstracts that fits the analysis stated in the title or mentioned in the abstract methods. Same in the first paragraph of the discussion, where the authors write "we performed a large-scale multi-trait POAG GWAS and identified 312 independent SNPs". This is even more misleading because the 312 were identified by "single-trait" POAG GWAS meta-analysis of diverse ancestries / and not by multi-trait analyses. The authors should revise the description of main analyses throughout the manuscript.

2. Sample compared to previous work

The whole work builds upon three previous glaucoma GWAS publications from overlapping authors: 1) Craig et al Nat Genet 2020, a MTAG multi-trait analysis on Europeans incorporating GWAS on VCDR ($n \sim 90K$), IOP ($n \sim 133K$) and glaucoma ($\sim 8K$ cases, $\sim 120K$ ctrls); 2) Gharahkhani et al Nat Commun 2021, a multi-ancestry GWAS meta-analysis for POAG combining POAG GWAS in Europeans ($\sim 24K$ cases, $\sim 307K$ ctrls), Asians ($\sim 7K$ cases, $\sim 40K$ ctrls) and Africans ($\sim 3K$ cases, $\sim 3K$ ctrls), and 3) Han et al AJHG 2021, a GWAS on VCDR estimated from a convolutional neuronal net on fundus images from UKB and the Canadian Longitudinal Study on Aging ($n \sim 115K$). The neuronal net estimation of VCDR was used in the current work to derive VCDR measurements as well. The study design section in the current work states European GWAS for POAG ($\sim 30K$ cases, $\sim 350K$ ctrls), VCDR ($\sim 111K$) and IOP ($\sim 153K$) as well as GWAS for POAG in Asians ($\sim 7K$ cases, $\sim 40K$ ctrls) and Africans ($\sim 3K$ cases, $\sim 3K$ ctrls). It seems that there is only a minor increase in sample size (e.g., 6K POAG cases more in EUR, no additional samples in non-EUR). The discussion states that the current work "tripled number of POAG loci". How can that be true given the minor increase in sample size? Please clarify the novelty of your sample compared to the existing work. Please provide descriptive statistics for your sample.

3. Novel loci compared to previous work

Same as for the analysed sample, I had difficulties following up on the novelty of loci. As mentioned before, the authors claim 312 identified loci (>100 novel), "tripled number of POAG loci". However they only compared their identified loci to the work by Craig et al Nat Genet 2020 and Gharahkhani et al Nat Commun 2021; but NOT to Han et al AJHG 2021. How can this be justified given that Han et al shows 230 independent lead SNPs for VCDR in their Table S4? How do the current results compare to those loci? The VCDR-specific loci, derived by clustering analyses, and the underlying neuroprotective drug targets, are the most important finding in the current work. This should not be criticized or down-weighted. This is relevant. But please clarify the novelty of loci in the current work.

4. Gene prioritization

The authors claim that they identify "69 drug targets for POAG" (row 234). While all of the 69 are indeed drug targets (for other disease), additional information is required to label them as "drug target for POAG". Otherwise, this is overstating the results. Especially because many of them are not supported by any mapping criteria. The genes highlighted in Table 1 are interesting but require some more discussion for the fact that additional evidence will be required to improve them into valuable POAG drug targets. For example, some genes are only supported by being the "Nearest gene" to the lead variant and lack any other mapping criteria (TWAS, pQTL, MAGMA, etc).

In addition, I have the following minor issues/questions:

1. How many of the novel loci replicated?
2. Why did the authors choose a $maf > 0.01$ threshold despite the large sample size? Are the novel loci less frequent?
3. The authors conduct MAGMA pathway analyses. Given the low genetic correlation between VCDR and IOP, shouldn't the pathway analyses be conducted based on VCDR- and/or IOP-specific loci to identify VCDR- and/or IOP-specific pathways?
4. I am missing some standard quality control metrics, e.g. GC lambdas, LD score estimates, QQ plots
5. Supp Table 1, and 2: There are many redundant columns (multiple chr/pos and allele columns) that make it difficult to follow the tables; please add allele frequencies; please align all effect sizes and frequencies to one reference allele column (preferably the risk-increasing allele) and omit the repeated allele columns.
6. Legends and footnotes with descriptions of the headers should be improved to give more detail in all Supp tables.
7. Multi-ancestry analyses: are there ancestry-specific POAG variants?
8. How do the coloc analyses handle multiple independent SNPs at individual loci?

Author Rebuttal to Initial comments

NG-A59653R: Large scale multi-trait genome-wide association analysis identifies hundreds of glaucoma risk loci

Reviewers' Comments:

Reviewer #1:

Remarks to the Author:

Han and coauthors conducted large-scale multi-trait analyses and meta-analyses and identified novel genetic loci associated with POAG. They also prioritized possible causal

genes and drug targets for POAG by querying a number of omics datasets. The authors further carried out MR analysis. There are several strengths, e.g. the sample size is impressive and the data analysis is extensive. This manuscript reminded me of the Craig et al. 2020 Nat Genet paper (from the same group) that I reviewed previously. This manuscript follows a similar design, i.e. MTAG, but with more samples from CLSA (1,358/16,455), MGB (1,415/18,632) and IGGC (not clear how many more samples than their previous meta-analyses). Despite the novel loci identified and the bioinformatics queries on potential causal genes and drug targets, the translational implications of these loci are still not clear.

1. With more samples included, it is likely that we will identify more loci with weaker effect sizes. However, how do these new loci of much weaker effect sizes contribute to diagnosis and treatment? Do they further improve the prediction of POAG, and by how much?

We thank the reviewer for acknowledging our efforts to conduct a more powerful POAG GWAS. In our current study, we included POAG GWAS with 29,241 cases and 350,181 controls of European ancestry, and its two key endophenotypes VCDR (N=111,724) and IOP (N=153,604). The studies are summarized in the New Supplementary Table 1 below. This leads to a large increase in power relative to either our 2020 analysis (~3 times as many glaucoma cases in case-control analysis, 20,122 additional IOP samples and 20,785 additional VCR samples, with a much more accurate VCDR phenotype via a machine-learning approach) or to our 2021 analysis (current study has 12,564 more POAG cases compared with Stage 1 in Gharahkhani et al. Nat Commun 2021, or 5,278 more POAG cases of European ancestry compared with Stage 1 + Stage 2 in Gharahkhani et al. Nat Commun 2021, but crucially IOP and VCDR were not used in Gharahkhani et al Nat

Commun 2021, leading to similar power in the 2020 and 2021 studies, with a similar number of loci discovered in each). One established method¹ for benchmarking the power of our new study against the previous one (previous IGGC POAG GWAS 16,677 POAG cases and 199,580 controls in stage 1 of European ancestry)² is to compute mean chi-squared value; this increases the mean value from 1.18 to 1.55, indicating we have tripled the effective sample size for POAG in the current MTAG GWAS.¹ As a result, in our European-only analysis, the number of independent SNPs increased to 263. With all ancestries included, the number of genome-wide significant SNPs increased beyond 300 (with the majority clearly replicated in our very large independent replication set (23andMe)).

We agree with the reviewer that, as expected, the novel loci had smaller effect sizes, although as we identified so many new loci, the amount of heritability explained markedly increased. In our current study, the proportion of the familial risk explained by the genome-wide significant independent SNPs was 14.1%. This represents a 50% increase over and above the previously reported estimate (9.4%), based on the previous IGGC meta-analysis that identified 127 genome-wide significant SNPs. An increase in variance explained is likely to lead to an increase in predictive accuracy although full evaluation of a new prediction algorithm is a full study in and of itself (certainly too much to fit into the current, already substantial, manuscript) and we are currently pursuing this as a separate manuscript. We cover the issue of how these novel loci may contribute to new/improved treatments for glaucoma in the response to the next question.

In the revision, we have now added the following Supplementary Table 1 and accompanying text.

New Supplementary Table 1. Summary of studies.

POAG MTAG in European	Trait	Study	Sample size	Note¹
	POAG	IGGC	15,229 POAG cases, 177,473 controls	Excluding UKB samples
		UKB	11,239 glaucoma cases, 13,7621 controls	New analysis (~ 3,000 more glaucoma cases)
		CLSA	1,358 glaucoma cases, 16,455 controls	New data
		MGB Biobank	1,415 glaucoma cases, 18,632 controls	New data
		-	Total: 29,241 cases, 350,181 controls	
	VCDR	UKB	68,240	New AI phenotyping, complete data including both eyes and both visits (baseline and follow-up)
		CLSA	18,304	New data, AI phenotyping
		IGGC	25,180	
		-	Total: 111,724	
	IOP	UKB	103,914	
		CLSA	18,421	New data
		IGGC	31,269	
		-	Total: 153,604	
Cross-ancestry meta-analysis	POAG	POAG MTAG in European	See above	All new data see above
		IGGC (Asian)	6,935 cases and 39,588 controls	
		IGGC (African)	3,281 cases and 2,791 controls	
Replication in 23andMe	POAG	23andMe	84,910 glaucoma cases and 2,736,075 controls	New data, with the largest sample size for replication

¹ New data: not presented in Craig et al Nat Genet 2020 or Gharahkhani et al Nat Commun 2021.

In our current study, we included POAG GWAS with 29,241 cases and 350,181 controls of European ancestry, and its two key endophenotypes VCDR (N=111,724) and IOP (N=153,604). This leads to a large increase in power relative to either our 2020 analysis (~3 times as many glaucoma cases in case-control analysis, 20,122 additional IOP samples and 20,785 additional VCR samples, with a much more accurate VCDR phenotype via a machine-learning approach) or to our 2021 analysis (current study has 12,564 more POAG cases compared with Stage 1 in Gharahkhani et al. Nat Commun 2021, or 5,278 more POAG cases of European ancestry compared with Stage 1 + Stage 2 in Gharahkhani et al. Nat Commun 2021, but crucially IOP and VCDR were not used in Gharahkhani et al Nat Commun 2021, leading to similar power in the 2020 and 2021 studies, with a similar number of loci discovered in each). One established method (Turley et al. Nature Genetics 2018) for benchmarking the power of our new study against the previous one (previous IGGC POAG GWAS 16,677 POAG cases and 199,580 controls in stage 1 of European ancestry) is to compute the mean chi-squared value; the mean value increases from 1.18 to 1.55, indicating we have tripled the effective sample size for POAG in the current MTAG GWAS.

2. With respect to drug targets, many pharmaceutical companies, including the cited Regeneron's WES paper, are using rare variants with large effect sizes.

Common variants

typically have smaller effect sizes. Furthermore, among the 312 POAG loci identified, only about 25 (8%) are included in Table 1, prioritized drug targets. What about the rest of the majority of POAG loci? How do these results translate to individual patients, who may have different causes from the 312 POAG loci?

We agree with the reviewer that common variants identified from GWAS typically have small effect sizes. We also agree with the reviewer that WES is an important complementary study design to identify rare variants with large effects,³ representing a small fraction of all human genetic variations that are most readily interpretable and actionable.³ However, many disease-causing variants are located in non-coding portions

of the genome, and WES will not typically provide information on these variants.^{4,5} The common variants harbored in the non-protein-coding region contains abundant information that can regulate gene expression, eg. long-non-coding RNAs (lncRNAs) and miRNAs.⁶ In particular, for common diseases and traits, many common genetic variants with small effects can contribute to the phenotype, known as polygenicity.⁷ For GWAS analysis of common variants, many loci have been well- replicated or have experimental validations, suggesting that they are genuine associations.⁸ While the effect sizes of GWAS signals may be modest, they still play an important role in identifying functional regulation genes and drug targets. Crucially, *a small effect size for a naturally occurring genetic variant does not mean modulating the target via a drug will have a small effect* (the classic example here is the HMGCR gene where GWAS-implicated variants have small effect on cholesterol levels and yet modulating HMGCRs-related enzyme with statins greatly alters cholesterol levels). In general, identifying the causal genetic variants and revealing the underlying biological mechanisms remain a critical bottleneck.^{7,8} In our current study, we have identified 312 POAG loci, of which 25 (8%) were prioritized based on different lines of genetic evidence using multi-omics datasets - we chose to highlight these as the “low hanging fruit” which are most likely to be promising drug targets. Naturally the hundreds of additional loci may provide unique biological insights too but, in the interest of space, we focused on only 25.

As to the translation to individual patients, as noted above, we hope our work facilitates development of new drugs to treat glaucoma. Current therapies for glaucoma, all of which aim to modulate IOP, are effective in many patients with glaucoma, not just those with very high pressure. Nevertheless, it is anticipated that as novel drug targets are identified, such as those with a neuroprotective effect, these would benefit many patients, regardless of the specific set of genetic variants they carry.

We have now added the following sentences in the Discussion section:

“As to the translation to individual patients, it is anticipated that as novel drug targets are

identified, such as those with a neuroprotective effect, these would benefit many patients, regardless of the specific set of genetic variants each individual harbors.”

3. The authors also investigated potential causal genes. However, without the support from wet-lab experiments, it is challenging to pin down causal genes/variants.

We agree with the reviewer that it is challenging to reveal causal genes and to delineate the biological mechanisms. In our current study, we have integrated a multitude of genetics and multi-omics datasets and used statistical approaches to prioritize causal genes for experimental follow up. To help readers identify the most relevant findings, we have tabulated some of the scenarios (table 1) where we have used multiple approaches to try to identify the potential causal genes. Pursuing further wet-lab experiments to investigate each of the novel loci in detail is important, but frankly a colossal amount of work and well beyond the scope of a single study.

We have now added the following sentences in the Discussion section:

“More detailed functional experiments are warranted to delineate the biological mechanisms of each of the identified loci.”

4. There is some inconsistency in the data analysis. For example, the authors used linear mixed models in their UKBB VCDR analysis (assuming related subjects were included) but required unrelated samples for their UKBB glaucoma analysis. There are methods, e.g. SAIGE, that can handle related subjects in case-control glaucoma analysis. The authors used ten genetic principal components (PCs) in their UKBB analysis but only used 5 PCs in their 23andMe analysis. The 23andMe dataset with

millions of samples is likely to be more heterogeneous than the UKBB dataset and ideally more PCs should be used.

We apologize for the inaccurate description. In our UKB glaucoma GWAS analysis, we indeed used SAIGE, and we have now updated the description as: “In our association analysis, we kept 11,239 glaucoma cases and 137,621 controls of European ancestry. We ran generalized mixed models in SAIGE (version 0.29.6) and adjusted for age, sex, and the first ten genetic principal components.”.

The 23andMe replication association analysis was performed independently. We also tested the covariates in 23andMe (see below Review Table 1): for the first five PCs, only PC1 and PC2 were associated with glaucoma at $P < 0.01$ (Bonferroni correction, $0.05/5$). Therefore, we think including more PCs would have a minimal effect on the association results.

Review Table 1. Covariates model information in 23andMe.

Covariate	Estimate	Std. Error	z value	Pr(> z)
Age	0.07156	0.000255	280.4	0
Sex (Female)	0.1181	0.007242	16.3	8.7e-60
PC1	-0.05411	0.003388	-16	2.0e-57
PC2	0.02165	0.003682	5.9	4.1e-09
PC3	-0.00304	0.00345	-0.9	0.38
PC4	-0.00745	0.003282	-2.3	0.023
PC5	0.0027	0.003452	0.8	0.43
Platformv3.0	0.14088	0.026969	5.2	1.8e-07
Platformv3.1	0.15521	0.019961	7.8	7.5e-15
Platformv4	-0.02056	0.008282	-2.5	0.013

5. Comparing to the very recent Gharahkhani et al (2021) meta-analysis with 34,179 cases and 349,321 controls from many of the same authors, what are the reasons that this meta-analysis of less samples of 29,241 cases and 350,181 controls identified many more loci than the previous very similar cross-ancestry meta-analysis and replication in 23andMe?

In the recent Gharahkhani et al (2021) meta-analysis², we included 16,677 cases and 199,580 controls of Europeans ancestry (stage 1), 6,935 cases and 39,588 controls of Asian ancestry, 3,281 cases and 2,791 controls of African ancestry, and 7,286 self-reported cases and 107,362 controls of European ancestry from UKB (stage 2).

In our current MTAG POAG analysis, for participants of European ancestry, we included POAG (29,241 cases and 350,181 controls) and its two key endophenotypes VCDR (N=111,724) and IOP (N=153,604). In the cross-ancestry analysis in the current study, we further included Asian population (6,935 cases and 39,588 controls) and African population (3,281 cases and 2,791 controls).

Overall, to compare the sample size of European ancestry, the main power was from **additional endophenotypes samples** (VCDR N=111,724 and IOP N=153,604) and **additional POAG cases** (12,564 more POAG cases compared with Stage 1 in Gharahkhani et al Nat Commun 2021, or 5,278 more POAG cases of European ancestry compared with Stage 1 + Stage 2 in Gharahkhani et al Nat Commun 2021). In our evaluation, compared with the previous IGGC POAG GWAS (with 16,677 POAG cases and 199,580 controls of European ancestry)², the value of the mean chi-square increased from 1.18 to 1.55, indicating we have **tripled the effective sample size** for POAG in the current MTAG GWAS,¹ providing an important advance over previous work

to identify more loci. Most of the novel loci replicated in a large independent cohort (23andMe), validating our approach.

Reviewer #2:

Remarks to the Author:

Han et al. describe results from multi-trait and multi-ancestry GWAS for glaucoma. Their multi-trait GWAS utilizes a well-established multi-trait framework (MTAG). Compared to single-outcome GWAS, MTAG yields higher power by exploiting the correlation between traits. The authors incorporate three glaucoma outcomes in MTAG: (i) case/control data on primary open-angle glaucoma (POAG; most common form of glaucoma; cases defined via electronic health record or self-report), (ii) intraocular pressure (IOP; known quantitative risk factor for glaucoma; measured via corneal-compensated IOP measurements) and (iii) vertical cup-disc ratio (VCDR, quantifies optic nerve head damage, known marker for glaucoma; assessed via convolutional neural net models trained on optical nerve head photographs). MTAG analyses were based on Europeans-only. In comparison, their multi-ancestry GWAS (adding data from Asian and African populations) were only conducted for POAG: They conducted single-outcome GWAS on POAG separately in Europeans, Asians and Africans and then combined ancestry-specific results via fixed-effect meta-analyses. They did not employ a multi-trait framework for the multi-ancestry analyses. Analyses are based on data from UK Biobank, Canadian Longitudinal Study on Aging, Mass General Brigham Biobank and International Glaucoma Genetic Consortium. This is an appropriate and powerful sample for this kind of study.

Their MTAG analyses based on Europeans identified 263 independent glaucoma-associated loci, of which 81 are claimed to be novel for glaucoma. Their multi-ancestry POAG GWAS identified 312 independent loci, of which 102 are claimed to be novel for

glaucoma. Importantly, the authors provide impressive replication of their findings in a large data set from 23andMe with ~80% of identified loci being replicated at a robust Bonferroni-corrected alpha-level.

The authors provide an impressive amount and state-of-the-art multi-omics GWAS follow-up analyses to prioritize genes at the identified loci. These include analyses of eQTL and sQTL colocalization (GTEx and retina tissue), transcriptome-wide association (in retina tissue), plasma proteome based on Mendelian Randomization. They identify 69 genes that are known drug targets genes for other diseases and claim them as "drug targets for POAG". Among the 69, the authors highlight 17 genes that were supported by at least two lines of

post-GWAS evidence and 10 that were located in VCDR-specific loci. The latter are claimed to be potential targets for neuroprotective treatments that may act through optic nerves, which are unavailable to date for glaucoma (all established treatments are on IOP). This is a relevant result. Finally, the authors found several complex traits/disease genetically correlated with glaucoma by LD score regression and Mendelian randomization analyses.

No fine-mapping

of causal variant resolution was performed. While the analyses are impressive and the results (especially the neuroprotective drug candidates) are highly relevant for glaucoma drug development, I have the following major concerns:

We thank the reviewer for the accurate summary of our work, and the very positive comments highlighting the advances of our findings.

1. Description of main analyses

The title states "multi-trait GWAS identified hundreds of glaucoma loci". The abstract

methods state "we conducted ... multi-trait GWAS". Yet, multi-trait GWAS results are not described in the abstract. Instead, results from a multi-ancestry meta-analysis of POAG (= single trait) are presented in the abstract: The authors write "Employing a multi-ancestry approach, we increased the number of independent risk loci to 312". This is misleading because the 312 were not identified from a multi-trait analysis. I expect to read the main result in the abstracts that fits the analysis stated in the title or mentioned in the abstract methods. Same in the first paragraph of the discussion, where the authors write "we performed a large-scale multi-trait POAG GWAS and identified 312 independent SNPs". This is even more misleading because the 312 were identified by "single-trait" POAG GWAS meta-analysis of diverse ancestries / and not by multi-trait analyses. The authors should revise the description of main analyses throughout the manuscript.

We thank the reviewer for the comments to improve our manuscript.

We apologize for not being sufficiently clear on this. In our study, we first performed a multi-trait POAG GWAS in participants of European ancestry, which dramatically boosted the power to identify glaucoma loci. The multi-trait POAG GWAS output was then meta-analyzed with GWAS data from participants of Asian and African ancestries. To make this clearer, we have now revised the manuscript as described below:

The relevant part of the abstract now reads:

"Since much of glaucoma heritability remains unexplained, we conducted a large-scale multi-trait GWAS in participants of European ancestry that combines POAG and these two associated traits (total subject sample size >600,000) to substantially improve genetic discovery power (263 loci). We further increased our power by then employing a multi-ancestry approach, which increased the number of independent risk loci to 312, with the vast majority replicating in a large independent cohort from 23andMe, Inc (total subject sample size >2.8 million; 296 loci replicated at $P < 0.05$, 240 after Bonferroni

correction).”

The relevant part of the discussion section now reads:

“In this study, we performed a large-scale multi-trait POAG GWAS identifying 263 loci in the single largest ancestry group (European ancestry). Additional cross-ancestry meta-analysis identifying 312 loci, including 109 which were completely distinct from previously reported loci.”

2. Sample compared to previous work

The whole work builds upon three previous glaucoma GWAS publications from overlapping authors: 1) Craig et al Nat Genet 2020, a MTAG multi-trait analysis on Europeans incorporating GWAS on VCDR (n~90K), IOP (n~133K) and glaucoma (~8K cases, ~120K ctrls); 2) Gharahkhani et al Nat Commun 2021, a multi-ancestry GWAS meta-analysis for POAG combining POAG GWAS in Europeans (~24K cases, ~307K ctrls), Asians (~7K cases, ~40K ctrls) and Africans (~3K cases, ~3K ctrls), and 3) Han et al AJHG 2021, a GWAS on VCDR estimated from a convolutional neuronal net on fundus images from UKB and the Canadian Longitudinal Study on Aging (n~115K). The neuronal net estimation of VCDR was used in the current work to derive VCDR measurements as well. The study design section in the current work states European GWAS for POAG (~30K cases, ~350K ctrls), VCDR (~111K) and IOP (~153K) as well as GWAS for POAG in Asians (~7K cases, ~40K ctrls) and Africans (~3K cases, ~3K ctrls). It seems that there is only a minor increase in sample size (e.g., 6K POAG cases more in EUR, no additional samples in non-EUR). The discussion states that the current work "tripled number of POAG loci". How can that be true given the minor increase in sample size? Please clarify the novelty of your sample compared to the existing work. Please provide descriptive

statistics for your sample.

We thank the reviewer for this comment and for giving us the opportunity to clarify the study design in our manuscript.

As presented above (responses to Reviewer #1, comments 1 and 5, New Supplementary Table 1), in the current MTAG POAG analysis, for participants of European ancestry, we included POAG (29,241 cases and 350,181 controls) and its two key endophenotypes VCDR (N=111,724) and IOP (N=153,604).

Compared with the previous IGGC POAG GWAS (Gharahkhani et al Nat Commun 2021), for participants of European ancestry, we have **additional endophenotype samples** (in total VCDR N=111,724 and IOP N=153,604) and **POAG cases** (12,564 more POAG cases compared with Stage 1 in Gharahkhani et al Nat Commun 2021, or 5,278 more POAG cases of European ancestry compared with Stage 1 + Stage 2 in Gharahkhani et al Nat Commun 2021). Compared with the previous IGGC POAG GWAS of European ancestry (with 16,677 POAG cases and 199,580 controls)², the values of mean chi-square increased from 1.18 to 1.55, indicating we have **tripled the effective sample size** for POAG in the current MTAG GWAS.¹

To give an intuitive example of the boosted effect sample size from our multi-trait analysis, in our previous projection work⁹, “we estimate that approximately 4 IOP samples or 7 VCDR samples contribute the same power as 1 sample in POAG GWAS (assuming a 1:1 ratio of case and control)”. The IOP and VCDR samples contributed approximately $(153,604/4 + 111,724/7)/2 = 27,181$ cases and the same number of controls (assuming a 1:1 ratio of case and control), which is similar to the estimation based on the increase of mean chi-square.

Compared with Craig et al Nat Genet 2020, the current study had ~3 times as many glaucoma cases in case-control analysis, ~20,000 additional IOP samples, and ~20,000 additional VCR samples, with a much more accurate VCDR phenotype via a machine-

learning approach.

For Han et al AJHG 2021, we only performed a VCDR GWAS.

In the revision, we have now added a new Supplementary Table 1 which helps the reader see where the increases in sample size are (see also our response to reviewer 1 who raised similar concerns).

3. Novel loci compared to previous work

Same as for the analysed sample, I had difficulties following up on the novelty of loci. As

mentioned before, the authors claim 312 identified loci (>100 novel), "tripled number of POAG loci". However they only compared their identified loci to the work by Craig et al Nat Genet 2020 and Gharahkhani et al NatCommun 2021; but NOT to Han et al AJHG 2021. How can this be justified given that Han et al shows 230 independent lead SNPs for VCDR in their Table S4? How do the current results compare to those loci? The VCDR-specific loci, derived by clustering analyses, and the underlying neuroprotective drug targets, are the most important finding in the current work. This should not be criticized or down-weighted. This is relevant. But please clarify the novelty of loci in the current work.

We thank the reviewer for this comment.

In the previous study (Han et al AJHG 2021), we used an automated AI labeling of the optic nerve head to perform GWAS analysis for vertical cup-to-disc ratio (VCDR) and vertical disc diameter (VDD). However, Han et al 2021 did not conduct a glaucoma GWAS. In our previous work^{10,11}, we have shown the genetic correlation between VCDR and POAG is 0.5 and only a subset of VCDR significant loci are associated with POAG. In the current study the focus is on **POAG loci**, and hence we only compared the

identified loci to the previous glaucoma GWAS work (Craig et al Nat Genet 2020 and Gharahkhani et al NatCommun 2021), rather than to Han et al AJHG 2021.

We have now also plotted all the 312 POAG SNPs from multi-ancestry meta-analysis against their effect sizes in VCDR and IOP grouped by different P value thresholds. As shown below (New Supplementary Figure 7 below), many POAG SNPs are not associated with VCDR, and similarly many POAG SNPs are not associated with IOP.

New Supplementary Figure 7: Comparison of the effect sizes for 312 POAG genome-wide significant independent SNPs from multi-ancestry meta-analysis against their effect sizes in VCDR and IOP grouped by different P value thresholds.

the comparison with VCDR. The x-axis shows the effect sizes in multi-ancestry meta-analysis of POAG. The y-axis shows the effect sizes in VCDR. The SNPs are shown in different colors based on different P values in VCDR ($P < 5e-8$, " $<0.05/312$ ": $5e-8 \leq P < 0.05/312$, " <0.05 ": $0.05/312 \leq P < 0.05$, and $P \geq 0.05$). Panel B shows the comparison with IOP. The SNPs are shown in different colors based on different P values in IOP ($P < 5e-8$, " $<0.05/312$ ": $5e-8 \leq P < 0.05/312$, " <0.05 ": $0.05/312 \leq P < 0.05$, and $P \geq 0.05$).

Gene prioritization

The authors claim that they identify "69 drug targets for POAG" (row 234). While all of the 69 are indeed drug targets (for other disease), additional information is required to label them as "drug target for POAG". Otherwise, this is overstating the results. Especially because many of them are not supported by any mapping criteria. The genes highlighted in Table 1 are interesting but require some more discussion for the fact that additional evidence will be required to improve them into valuable POAG drug targets. For example, some genes are only supported by being the "Nearest gene" to the lead variant and lack any other mapping criteria (TWAS, pQTL, MAGMA, etc).

We thank the reviewer for this suggestion. From our POAG GWAS, we found 69 POAG associated genes were drug targets under clinical trials for eye diseases and many other

diseases. Therefore, we prioritized the 69 gene list as *potential* drug targets for POAG.

In our Table 1, we included 17 genes with at least two levels of genetic evidence and 8 genes that affect POAG most likely through VCDR, without an apparent effect on IOP. We agree with the reviewer that the genetic evidence from our study needs additional functional experiments and clinical trials to support the findings. We have now added the following sentences in the Discussion section:

"Finally, we prioritize a list of potential drug targets for POAG based on genetic evidence;

additional functional experiments and clinical trials are needed to support these findings.”

In the Results section, we also modified “we identified 69 drug target genes for POAG” to instead read “we identified 69 potential drug target genes for POAG”

In the Discussion, we also modified “Leveraging omics data and multiple levels of genetic evidence, we prioritized 69 putative drug targets for POAG” to instead read “Leveraging omics data and multiple levels of genetic evidence, we prioritized 69 putative drug targets for POAG (including 17 with at least two levels of supporting genetic evidence)”.

In addition, I have the following minor issues/questions:

1. How many of the novel loci replicated?

We thank the reviewer for this suggestion. We have now added a new supplementary figure 2 and have shown the replication rates of novel loci at different P value thresholds.

For the 81 previously unknown genome-wide significant independent SNPs identified from the POAG multi-trait GWAS in European ancestry, the replication rates in an independent cohort using 23andMe were: 38% SNPs (N=31) passed the genome-wide significance level ($P < 5 \times 10^{-8}$) in the 23andMe study, 73% SNPs (N=59) were significant after Bonferroni correction ($P < 0.00062$), and 96% SNPs (N=78) reached a nominal significance level ($P < 0.05$). For the novel loci, we found a very high concordance of the effect sizes between the MTAG discovery and the 23andMe replication (Pearson's coefficient 0.94, $P = 1.42 \times 10^{-38}$).

³⁸).

For the 109 previously unknown genome-wide significant independent SNPs identified from POAG multi-ancestry meta-analysis, the replication rates in an independent cohort using 23andMe were: 38% SNPs (N=39) passed the genome-wide significance level ($P < 5 \times 10^{-8}$) in the 23andMe study, 66% SNPs (N=68) were significant after Bonferroni correction ($P < 0.0005$), and 96% SNPs (N=99) reached a nominal significance level ($P < 0.05$). For the novel loci, we found a very high concordance of the effect sizes between the MTAG discovery and the 23andMe replication (Pearson's coefficient 0.939, P value= 1.57×10^{-48}).

New Supplementary Figure 2: Comparison of the effect sizes for genome-wide significant independent SNPs by known loci and previously known loci. Panel A shows effect sizes for known genome-wide significant independent SNPs identified from the POAG multi-trait GWAS in European ancestry versus glaucoma GWAS in 23andMe. Panel B shows 81 previously unknown genome-wide significant independent SNPs identified from the POAG multi-trait GWAS in European ancestry versus glaucoma GWAS in 23andMe. The Pearson's coefficient is 0.94 (P value= 1.42×10^{-38}). Panel C

shows effect sizes for known genome-wide significant independent SNPs identified from the POAG multi-ancestry meta-

analysis versus glaucoma GWAS in 23andMe. Panel D shows 109 previously unknown genome-wide significant independent SNPs identified from POAG multi-ancestry meta-analysis versus glaucoma GWAS in 23andMe. The Pearson's coefficient is 0.939 (P value= 1.57×10^{-48}).

For the 81 previously unknown genome-wide significant independent SNPs identified from the POAG multi-trait GWAS in European ancestry, the replication rates in an independent cohort using 23andMe were: 38% SNPs (N=31) passed the genome-wide significance level ($P < 5 \times 10^{-8}$) in the 23andMe study, 73% SNPs (N=59) were significant after Bonferroni correction ($P < 0.00062$), and 96% SNPs (N=78) reached a nominal significance level ($P < 0.05$).

For the 109 previously unknown genome-wide significant independent SNPs identified from POAG multi-ancestry meta-analysis, the replication rates in an independent cohort using 23andMe were: 38% SNPs (N=39) passed the genome-wide significance level ($P < 5 \times 10^{-8}$) in the 23andMe study, 66% SNPs (N=68) were significant after Bonferroni correction ($P < 0.0005$), and 96% SNPs (N=99) reached a nominal significance level ($P < 0.05$).

2. Why did the authors choose a maf>0.01 threshold despite the large sample size?

Are the novel loci less frequent?

We agree with the reviewer that with the large sample size in the UKB, the imputation accuracy will be high for most SNPs even with MAF at 0.1% (Figure 1e Halldorsson et al.¹²). However, there are two primary reasons we chose MAF at 1% in our current analysis. First, we included several cohorts with smaller sample sizes, and thus a MAF

at 1% is more appropriate. Second, in the multi-trait analysis using MTAG, only overlapping SNPs from different datasets are retained for downstream analysis, and the default filtering criteria is a MAF at 1%.

We also include a scatter plot (New Supplementary Figure 3 below) for minor allele frequency (MAF) and effect size (absolute value) for the novel loci. Similar to other traits, novel loci have smaller effect sizes compared to known loci. We did not observe a pattern that those novel loci are less frequent (only three SNPs with MAF < 0.05, vertical dash line).

New Supplementary Figure 3: Scatterplot of minor allele frequency (MAF) and effect size (absolute value) for the novel loci in the POAG MTAG analysis (“1” for novel SNPs; “0” for known SNPs).

3. The authors conduct MAGMA pathway analyses. Given the low genetic correlation between VCDR and IOP, shouldn't the pathway analyses be conducted based on VCDR- and/or IOP-specific loci to identify VCDR- and/or IOP-specific pathways?

We thank the reviewer for this suggestion. The strength of the multi-trait approach (MTAG) that we employ in the study is that it is based on multiple input traits (here POAG, VCDR and IOP) and that it produces an output result which is specific to the main trait of interest to us (POAG). In our situation, although the correlation between IOP and VCDR is fairly low, both are strongly correlated with POAG and both IOP and VCDR improve our power (with the validity of the large number of resultant loci confirmed in our independent 23andMe based replication). We repeated the multitrait (MTAG) analysis with only POAG and VCDR

included as input (leaving IOP out of MTAG); however, since there are fewer data contributing, the number of loci identified is smaller (similarly for POAG and IOP, leaving out VCDR). Hence to maximize our power to identify novel pathways, we conducted the pathway analysis using all three input traits, rather than just a subset. Although in principle we could include the pathway results for e.g. the MTAG model with POAG and VCDR as input, we would prefer not to do this because the results would still be likely to reflect the influence of IOP (since the POAG GWAS includes many genes which are also IOP genes). If one is interested in “pure” VCDR or IOP pathway results, these can be found in our previous studies which focused solely on VCDR¹¹ or solely on IOP¹³.

4. I am missing some standard quality control metrics, e.g. GC lambdas, LD score estimates, QQ plots

We thank the reviewer for this suggestion. We have now added a new supplementary Figure 4 to show the quality control metrics.

New Supplementary Figure 4. Quantile-quantile plots for POAG GWAS. The upper panel shows the quantile-quantile plot for multi-trait POAG GWAS in participants of European ancestry. The quantile-quantile plot is based on one million randomly selected SNPs. Linkage disequilibrium (LD) score regression intercept is used to assess the genomic inflation, the intercept is 0.957 (standard error, SE=0.013, attenuation ratio < 0), and the lambda value is 1.27. The lower panel shows the quantile-quantile plot for POAG cross-ancestry meta-analysis. The lambda value is 1.28. Because of the multiple ancestries included, LDSC was not performed.

5. Supp Table 1, and 2: There are many redundant columns (multiple chr/pos and allele columns) that make it difficult to follow the tables; please add allele frequencies; please align all effect sizes and frequencies to one reference allele column (preferably the risk-increasing allele) and omit the repeated allele columns.

We thank the reviewer for this suggestion. We have now updated the Supplementary Tables to add the effect allele frequencies and to remove the repeated allele columns.

6. Legends and footnotes with descriptions of the headers should be improved to give more detail in all Supp tables.

We thank the reviewer for this suggestion. We have now added a new row to describe all header columns in the Supplementary Tables.

7. Multi-ancestry analyses: are there ancestry-specific POAG variants?

We thank the reviewer for this comment. In our multi-ancestry analysis, we included

participants from European ancestry, Asian ancestry, and African ancestry. The sample size of non-European ancestry participants in our POAG GWAS was still relatively small. For instance, we included participants of Asian ancestry (6,935 cases and 39,588 controls) and African ancestry (3,281 cases and 2,791 controls), which were only a small fraction compared with the number of participants of European ancestry.

For European ancestry, we identified hundreds of loci and most are not genome-wide significant in the Asian & African ancestries alone. However, the main reason for this is almost certainly the limited power, due to much smaller sample size in those ancestries. We have shown that the overall concordance of SNP effect sizes was moderate to high between participants of European ancestry and Asian & African ancestry (Figures 3C and 3D), indicating most POAG variants identified from European ancestry might have an effect on POAG in participants of Asian & African ancestry.

For Asian & African ancestry, we previously reported 10 significant loci in Asian ancestry ($P < 5e-8$), all of which are known POAG loci.² In the POAG GWAS of African ancestry, one locus (rs16944405 within IQGAP1) reached the genome-wide significance level ($P = 3e-08$) and showed no association in POAG GWAS in European & Asian ancestry.²

8. How do the coloc analyses handle multiple independent SNPs at individual loci?

We thank the reviewer for the comment. In the traditional colocalization analysis in the

“COLOC” R package, the assumption is a single causal variant for each trait at individual loci (Giambartolomei et al. 2014, PLoS Genetics).¹⁴ A recent Sum of Single Effects (SuSiE) regression framework was adapted to colocalization analysis to account for multiple causal variants in a region (“coloc.susie” function, Wallace, 2021, PLoS Genetics)¹⁵.

As recommended by Dr Wallace¹⁵, single-COLOC can be useful when no credible sets can be detected with confidence by COLOC-SuSiE. We have now used a hybrid approach to use coloc-SuSiE where possible, but to use coloc-single when SuSiE cannot identify any credible sets¹⁵.

References:

1. Turley, P. *et al.* Multi-trait analysis of genome-wide association summary statistics using MTAG. *Nat. Genet.* **50**, 229–237 (2018).
2. Gharahkhani, P. *et al.* Genome-wide meta-analysis identifies 127 open-angle glaucoma loci with consistent effect across ancestries. *Nat. Commun.* **12**, 1258 (2021).
3. Szustakowski, J. D. *et al.* Advancing human genetics research and drug discovery through exome sequencing of the UK Biobank. *Nat. Genet.* 1–7 (2021).
4. Wong, A. K., Sealfon, R. S. G., Theesfeld, C. L. & Troyanskaya, O. G. Decoding disease: from genomes to networks to phenotypes. *Nat. Rev. Genet.* **22**, 774–790 (2021).
5. Przybyla, L. & Gilbert, L. A. A new era in functional genomics screens. *Nat. Rev. Genet.* **23**, 89–103 (2021).
6. Gil, N. & Ulitsky, I. Regulation of gene expression by cis-acting long non-coding RNAs. *Nat.*

Rev. Genet. **21**, 102–117 (2019).

7. Uffelmann, E. *et al.* Genome-wide association studies. *Nature Reviews Methods Primers* **1**, 1– 21 (2021).
8. Gallagher, M. D. & Chen-Plotkin, A. S. The Post-GWAS Era: From Association to Function.

Am. J. Hum. Genet. **102**, 717–730 (2018).

9. Han, X., Hewitt, A. W. & MacGregor, S. Predicting the Future of Genetic Risk Profiling of Glaucoma: A Narrative Review. *JAMA Ophthalmol.* **139**, 224–231 (2021).
10. Craig, J. E. *et al.* Multitrait analysis of glaucoma identifies new risk loci and enables polygenic prediction of disease susceptibility and progression. *Nat. Genet.* **52**, 160–166 (2020).
11. Han, X. *et al.* Automated AI labeling of optic nerve head enables insights into cross-ancestry glaucoma risk and genetic discovery in >280,000 images from UKB and CLSA. *Am. J. Hum. Genet.* **108**, 1204–1216 (2021).
12. Halldorsson, B. V. *et al.* The sequences of 150,119 genomes in the UK biobank. *bioRxiv* 2021.11.16.468246 (2021) doi:10.1101/2021.11.16.468246.
13. MacGregor, S. *et al.* Genome-wide association study of intraocular pressure uncovers new pathways to glaucoma. *Nat. Genet.* **50**, 1067–1071 (2018).
14. Giambartolomei, C. *et al.* Bayesian test for colocalisation between pairs of genetic association studies using summary statistics. *PLoS Genet.* **10**, e1004383 (2014).
15. Wallace, C. A more accurate method for colocalisation analysis allowing for multiple causal variants. *PLoS Genet.* **17**, e1009440 (2021)

Decision Letter, first revision:

23rd August 2022

Dear Xikun,

Your revised Article "Large scale multi-trait genome-wide association analysis identifies hundreds of glaucoma risk loci" has been seen by the original referees. You will see from their comments below that, while Reviewer #2 is now satisfied, Reviewer #1 has raised a few ongoing concerns. We remain interested in the possibility of publishing your study in Nature Genetics, but we would like to consider your response to these remaining concerns in the form of a further revision before we make a final decision on publication.

As before, to guide the scope of the revisions, the editors discuss the referee reports in detail within the team, including with the chief editor, with a view to identifying key priorities that should be addressed in revision, and sometimes overruling referee requests that are deemed beyond the scope of the current study. In this case, although we would not require you to construct and validate a new polygenic prediction model for glaucoma as part of the current study, we ask that you clarify how the estimate of heritability explained was calculated and that you provide further details on how the association analyses were performed. We again hope you will find this prioritized set of referee points to be useful when revising your study. Please do not hesitate to get in touch if you would like to discuss these issues further.

We therefore invite you to revise your manuscript again taking into account all reviewer and editor comments. Please highlight all changes in the manuscript text file. At this stage, we will need you to upload a copy of the manuscript in MS Word .docx or similar editable format.

*2) If you have not done so already please begin to revise your manuscript so that it conforms to our Article format instructions, available http://www.nature.com/ng/authors/article_types/index.html here. Refer also to any guidelines provided in this letter.

[redacted]

We again hope to receive your revised manuscript within 4-8 weeks. If you cannot send it within this time, please let us know.

Sincerely,
Kyle

Kyle Vogan, PhD
Senior Editor
Nature Genetics
<https://orcid.org/0000-0001-9565-9665>

Referee expertise:

Referee #1: Genetics, vision disorders, bioinformatics

Referee #2: Genetics, statistical methods, complex traits

Reviewers' Comments:

Reviewer #1:
Remarks to the Author:

Given the excellent reports from their previous studies, Craig et al. 2020 Nat Genet and Gharahkhani et al. 2021 (both of large sample sizes), using the same methodology, multi-trait and meta-analysis, the novelty of this manuscript is not clear.

It is not clear how much improvement the manuscript has made in comparison to their Craig et al. 2020 paper without providing the prediction piece. This should be straightforward to do, e.g. using their previous method, given their expertise. Compared to Gharahkhani et al. 2021, this study has 5,278 more POAG cases. Given the large authorship overlap, the similarity of the datasets, and the seemingly apparent advantage of multi-trait as the authors stated, it is not clear why the multi-trait was not used previously.

The authors used mean chi-squared value to justify the improvement over the previous studies of large sample sizes. In fact, to make it more apparent, $\chi^2 = n \cdot r^2$ ($df=1$), assuming a simple case, a binary trait and a SNP. With larger sample size, you will see more SNPs passing a fixed significance cutoff. However, the correlation detected is much smaller. It is challenging to be enthusiastic about this after their similar very large-sample studies were already reported. As n goes to infinity, shall we declare the majority of SNPs significant?

The authors mentioned that the amount of heritability explained markedly increased, from 9.4% to 14.1%. It is not clear how it was done and no details were provided. The heritability explained can be improved by increasing the variants used in estimation, despite whether they are genome-wide significant or not. Yang et al. 2010, 2015 Nat Genet gave excellent examples using all SNPs to estimate heritability.

SAIGE. Other than a brief citation, no details were provided on how it was used on so many cohorts.

There are over three million individuals in the 23andMe dataset. For such a large heterogeneous dataset, five PCs are not enough to cover the subpopulation space.

Reviewer #2:
Remarks to the Author:

I thank the authors for the extensive response and appreciate the addition of supplementary material. The authors clarified all of my concerns.

Author Rebuttal, first revision:

NG-A59653R1: Large scale multi-trait genome-wide association analysis identifies hundreds of glaucoma risk loci

Reviewers' Comments:

Reviewer #1:

Remarks to the Author:

Given the excellent reports from their previous studies, Craig et al. 2020 Nat Genet and Gharahkhani et al. 2021 (both of large sample sizes), using the same methodology, multi-trait and meta-analysis, the novelty of this manuscript is not clear.

It is not clear how much improvement the manuscript has made in comparison to their Craig et al. 2020 paper without providing the prediction piece. This should be straightforward to do, e.g. using their previous method, given their expertise. Compared to Gharahkhani et al. 2021, this study has 5,278 more POAG cases. Given the large authorship overlap, the similarity of the datasets, and the seemingly apparent advantage of multi-trait as the authors stated, it is not clear why the multi-trait was not used previously.

Response:

We thank the reviewer for the comments. A major strength of our study is that the effective sample size of our GWAS is three times larger than previous glaucoma GWASs. As stated in our previous response, compared to our Craig et al. 2020 study, we have shown the large increase of sample size in the Supplementary Table 1 (~3 times as many glaucoma cases in case-control analysis, 20,122 additional IOP samples and 20,785 additional VCR samples, with a much more accurate VCDR phenotype via a machine-learning approach).

RE: *“Compared to Gharahkhani et al. 2021, this study has 5,278 more POAG cases. Given the large authorship overlap, the similarity of the datasets, and the seemingly apparent advantage of multi-trait as the authors stated, it is not clear why the multi-trait was not used previously.”*

At the time the analysis was done for the Gharahkhani et al. 2021 study, some of the key items which improve the power in our current manuscript were simply not available:

- Firstly, the AI derived phenotyping for the UKB data was not yet available; these data markedly increase the effective sample size of our multitrait analysis (more accurate phenotype, larger number of individuals with a usable phenotype, most individuals had phenotype on both eyes rather than just one [as in Craig et al 2020]).
- Secondly, the CLSA data was not available to us. This data markedly improved power because in addition to contributing to case-control data, it also contributed to both IOP and

AI derived VCDR (~20,000 samples). For both IOP and VCDR the traits are measured at multiple time points which further improves power.

- Thirdly, the 23andMe replication set available to us was substantially larger than that available for the 2021 paper (84,910 cases and 2,736,075 controls now, versus 43,254 cases and controls 1,471,118 in the 2021 paper). The increased replication sample size was important when we sought to validate the novel SNPs uncovered by our expanded multi-trait analysis. With only 43,254 cases available in the replication set many of our novel SNPs would not have replicated simply due to limited statistical power.

For the prediction, we have shown the proportion of the familial risk explained by the genome-wide significant independent SNPs was 14.1% in the current study, which represents a 50% increase over and above the previously reported estimate (9.4%), based on the previous IGGC meta-analysis that identified 127 genome-wide significant SNPs. Following discussion with the editor, we plan to pursue a full study evaluating polygenic risk prediction in a separate manuscript.

The authors used mean chi-squared value to justify the improvement over the previous studies of large sample sizes. In fact, to make it more apparent, $\chi^2 = n \cdot r^2$ (df=1), assuming a simple case, a binary trait and a SNP. With larger sample size, you will see more SNPs passing a fixed significance cutoff. However, the correlation detected is much smaller. It is challenging to be enthusiastic about this after their similar very large-sample studies were

already reported. As n goes to infinity, shall we declare the majority of SNPs significant?

Response:

We used the mean chi-square approach (Turley et al. 2018) to benchmark the GWAS-equivalent sample size across different studies. The method was described in detail in the Section “GWAS-equivalent sample size for MTAG” by Turley et al 2018.

With even larger sample sizes, it is likely that more genetic signals will reach genome-wide significance although the number of loci and % genetic variance explained will eventually plateau. Whilst it is difficult to be certain when extrapolating from existing data, we previously published a statistical modeling of this for glaucoma (Han et al.¹, JAMA Ophthalmology 2021 - referenced in our manuscript, figure 2 from our 2021 paper is pasted below for your convenience); at effective sample sizes in the 50,000-200,000 range, steady increases in sample size are likely to lead to increases in number of loci/variance explained. When N reaches several hundreds of thousands, the benefits of ever larger GWASs are likely to diminish. In the meantime, our current paper robustly identifies many novel loci, providing new biological insights as well as helping prioritize potential drug targets.

Figure 2. Projection of the Number of Discovered Single-Nucleotide Variants (SNVs) and Genetic Variance Explained for Glaucoma, Intraocular Pressure (IOP), and Vertical Cup-Disc Ratio (VCDR)

The x-axis is the sample size of genome-wide association study summary statistics. For glaucoma, the sample size equals the total number of cases and controls, assuming a 1:1 ratio. Diamond symbols show the projection at different sample sizes (roughly current sample size, double, and quadruple). In Panel A,

the y-axis is the projected number of independent SNVs. In panel B, the y-axis is the genetic variance explained (%), which is equal to phenotypic variance explained multiply by heritability.

The authors mentioned that the amount of heritability explained markedly increased, from 9.4% to 14.1%. It is not clear how it was done and no details were provided. The heritability explained can be improved by increasing the variants used in estimation, despite whether they are genome-wide significant or not. Yang et al. 2010, 2015 Nat Genet gave excellent examples using all SNPs to estimate heritability.

Response:

We thank the reviewer for the comment. Some of the requested details were provided in the section “Define independent loci and novel POAG loci” although to better address the reviewer’s concern we have now added a new Section “Proportion of familial risk explained”, and it reads: “The proportion of familial risk explained was computed as the sum of $p^*(1-p)^*b^2/\log_e(9.2)$ over all independent genome-wide significant SNPs (as defined in “**Definition of independent loci and novel POAG loci**”), where p is the allele frequency, b the log odds ratio and 9.2 the increased risk in first degree relatives as estimated in a previous study.^{2,3}”

Regarding the reviewer’s comment about “Yang et al. 2010, 2015 Nat Genet”, the Yang work estimated the heritability based on all common SNPs. Crucially Yang’s work only estimates the collective effect of all common SNPs and it does not identify the specific SNPs of interest. Our GWAS does identify the specific SNPs of interest and our 14.1% figure shows that we have markedly increased the variance explained by the SNPs which reach a stringent level of statistical significance.

SAIGE. Other than a brief citation, no details were provided on how it was used on so many cohorts.

Response:

We have now added more details for SAIGE association analysis procedures. It now reads:

“We ran generalized mixed models in SAIGE (version 0.29.6)⁴ and adjusted for age, sex, and the first ten genetic principal components. In the SAIGE analysis, generalized linear mixed models with two steps were fitted to account for unbalanced case-control ratios and sample relatedness. The first step (fitNULLGLMM) was used to estimate variance component and model parameters. The second step (SPAGMMATtest) performed single variant score tests with saddlepoint approximation based on logistic mixed models.⁴”

There are over three million individuals in the 23andMe dataset. For such a large heterogeneous dataset, five PCs are not enough to cover the subpopulation space.

Response:

In a previous evaluation (Watanabe et al. Nature Genetics, 2022⁵), the first five PCs in 23andMe explained more variance than the first ten PCs in the UKB. Moreover, Watanabe et al.⁵ have shown that the variance in the 23andMe was flat after the fifth PC while this plateau was reached after the tenth PC in the UKB.

The detailed descriptions are available at

<https://www.nature.com/articles/s41588-022-01124-w#Sec6> and Supplementary Fig. 1

(pasted below for your convenience) by Watanabe et al.⁵

We have now added the following sentences in the Method section:

“Only the first 5 principal components were included as a previous study has shown that the first 5 principal components in the 23andMe dataset explained more variance than the first ten principal components in the UKB, and the total variance in the 23andMe reached a plateau after the fifth principal components while in the UKB the variance was flat after the tenth principal components.⁵”

Supplementary Figures

Supplementary Fig. 1. Total variance explained by genetic principal components. The principal components were computed for EUR samples of the 23andMe (a) and UKB (b).

Reviewer #2:

Remarks to the Author:

I thank the authors for the extensive response and appreciate the addition of supplementary material. The authors clarified all of my concerns.

Response:

We thank the reviewer for all the comments to help us improve our manuscript.

This email has been sent through the Springer Nature Tracking System NY-610A-NPG&MTS

References:

1. Han, X., Hewitt, A. W. & MacGregor, S. Predicting the Future of Genetic Risk Profiling of Glaucoma: A Narrative Review. *JAMA Ophthalmol.* **139**, 224–231 (2021).
2. Wolfs, R. C. *et al.* Genetic risk of primary open-angle glaucoma. Population-based familial aggregation study. *Arch. Ophthalmol.* **116**, 1640–1645 (1998).
3. Bahcall, O. Common variation and heritability estimates for breast, ovarian and prostate cancers. *Nat. Genet.* (2013) doi:10.1038/ngicogs.1.
4. Zhou, W. *et al.* Efficiently controlling for case-control imbalance and sample relatedness in large-scale genetic association studies. *Nat. Genet.* **50**, 1335–1341 (2018).
5. Watanabe, K. *et al.* Genome-wide meta-analysis of insomnia prioritizes genes associated with metabolic and psychiatric pathways. *Nat. Genet.* **54**, 1125–1132 (2022).

Decision Letter, second revision:

18th November 2022

Dear Xikun,

Thank you for submitting your revised manuscript "Large scale multi-trait genome-wide association analysis identifies hundreds of glaucoma risk loci" (NG-A59653R1). In light of the previous referee comments and your responses to Reviewer #1, we will be happy in principle to publish your study in

Nature Genetics as an Article pending final revisions to comply with our editorial and formatting guidelines.

We are now performing detailed checks on your paper, and we will send you a checklist detailing our editorial and formatting requirements soon. Please do not upload the final materials or make any revisions until you receive this additional information from us.

Thank you again for your interest in Nature Genetics. Please do not hesitate to contact me if you have any questions.

Sincerely,
Kyle

Kyle Vogan, PhD
Senior Editor
Nature Genetics
<https://orcid.org/0000-0001-9565-9665>

Final Decision Letter:

19th May 2023

Dear Xikun,

I am delighted to say that your manuscript "Large-scale multi-trait genome-wide association analyses identify hundreds of glaucoma risk loci" has been accepted for publication in an upcoming issue of Nature Genetics.

Your paper will be published online after we receive your corrections and will appear in print in the

next available issue. You can find out your date of online publication by contacting the Nature Press Office (press@nature.com) after sending your e-proof corrections. Now is the time to inform your Public Relations or Press Office about your paper, as they might be interested in promoting its publication. This will allow them time to prepare an accurate and satisfactory press release. Include your manuscript tracking number (NG-A59653R2) and the name of the journal, which they will need when they contact our Press Office.

Before your paper is published online, we will be distributing a press release to news organizations worldwide, which may very well include details of your work. We are happy for your institution or funding agency to prepare its own press release, but it must mention the embargo date and Nature Genetics. Our Press Office may contact you closer to the time of publication, but if you or your Press Office have any enquiries in the meantime, please contact press@nature.com.

Please note that Nature Genetics is a Transformative Journal (TJ). Authors may publish their research with us through the traditional subscription access route or make their paper immediately open access through payment of an article-processing charge (APC). Authors will not be required to make a final decision about access to their article until it has been accepted. [Find out more about Transformative Journals](https://www.springernature.com/gp/open-research/transformative-journals)

Authors may need to take specific actions to achieve [compliance](https://www.springernature.com/gp/open-research/funding/policy-compliance-faqs) with funder and institutional open access mandates. If your research is supported by a funder that requires immediate open access (e.g. according to [Plan S principles](https://www.springernature.com/gp/open-research/plan-s-compliance)), then you should select the gold OA route, and we will direct you to the compliant route where possible. For authors selecting the subscription publication route, the journal's standard licensing terms will need to be accepted, including [self-archiving-and-license-to-publish](https://www.nature.com/nature-portfolio/editorial-policies/self-archiving-and-license-to-publish). Those licensing terms will supersede any other terms that the author or any third party may assert apply to any version of the manuscript.

Please note that Nature Portfolio offers an immediate open access option only for papers that were first submitted after 1 January 2021.

To assist our authors in disseminating their research to the broader community, our SharedIt initiative provides you with a unique shareable link that will allow anyone (with or without a subscription) to

read the published article. Recipients of the link with a subscription will also be able to download and print the PDF.

If you have not already done so, we invite you to upload the step-by-step protocols used in this manuscript to the Protocols Exchange, part of our on-line web resource, natureprotocols.com. If you complete the upload by the time you receive your manuscript proofs, we can insert links in your article that lead directly to the protocol details. Your protocol will be made freely available upon publication of your paper. By participating in natureprotocols.com, you are enabling researchers to more readily reproduce or adapt the methodology you use. [Natureprotocols.com](http://natureprotocols.com) is fully searchable, providing your protocols and paper with increased utility and visibility. Please submit your protocol to <https://protocolexchange.researchsquare.com/>. After entering your nature.com username and password you will need to enter your manuscript number (NG-A59653R2). Further information can be found at <https://www.nature.com/nature-portfolio/editorial-policies/reporting-standards#protocols>

Sincerely,
Kyle

Kyle Vogan, PhD
Senior Editor
Nature Genetics
<https://orcid.org/0000-0001-9565-9665>